# Numb positively regulates Hedgehog signaling at the ciliary pocket

Xiaoliang Liu [1,10], Patricia T. Yam [2,10], Sabrina Schlienger[2,3], Eva Cai [1], Jingyi Zhang [1], Wei-Ju Chen[2,4], Oscar Torres Gutierrez[1], Vanesa Jimenez Amilburu[2], Vasanth Ramamurthy[2], Alice Y. Ting [5,6], Tess C. Branon [5,6,9], Michel Cayouette [2,3,7], Risako Gen [8], Tessa Marks [8], Jennifer H. Kong [8], Frédéric Charron [2,3,7] ✉ & Xuecai Ge [1] ✉

Hedgehog (Hh) signaling relies on the primary cilium, a cell surface organelle that serves as a signaling hub for the cell. Using proximity labeling and quantitative proteomics, we identify Numb as a ciliary protein that positively regulates Hh signaling. Numb localizes to the ciliary pocket and acts as an endocytic adaptor to incorporate Ptch1 into clathrin-coated vesicles, thereby promoting Ptch1 exit from the cilium, a key step in Hh signaling activation. Numb loss impedes Sonic hedgehog (Shh)-induced Ptch1 exit from the cilium, resulting in reduced Hh signaling. Numb loss in spinal neural progenitors reduces Shh-induced differentiation into cell fates reliant on high Hh activity. Genetic ablation of Numb in the developing cerebellum impairs the proliferation of granule cell precursors, a Hh-dependent process, resulting in reduced cerebellar size. This study highlights Numb as a regulator of ciliary Ptch1 levels during Hh signal activation and demonstrates the key role of ciliary pocket-mediated endocytosis in cell signaling.

The primary cilium serves as a cellular antenna that senses molecules and conveys their signal from the extracellular milieu into the cell[1,2]. Hedgehog (Hh) signaling was the first signaling pathway discovered to rely on the primary cilium[3]. Hh signal transduction is carried out by the dynamic transport of signal transducers in and out of the cilium. In the basal state, the Hh receptor Ptch1 resides in the cilium where it prevents the accumulation and activation of Smo in the cilium[4]. Upon Shh stimulation, Ptch1 exits the cilium. Reciprocally, Smo accumulates and is activated in the cilium. Active Smo triggers a signaling cascade that eventually activates Gli transcription factors which then turn on the transcription of Hh target genes[5–7]. During this process, Ptch1 exit from the cilium is a key step that initiates Hh signaling.

Recent studies suggest that Ptch1 acts as a sterol transporter to control local sterol components, such as cholesterols, in the cilium[8–12]. Binding to certain oxysterols is required for Smo activation[13,14]. Therefore, Ptch1 is believed to inhibit Smo by restricting Smo's access to activating sterols in the ciliary membrane. The clearance of Ptch1 from the cilium correlates with Shh-mediated pathway activation and is necessary for maximal activation of Hh signaling[4,15]. However, the endocytic machinery that mediates Ptch1 exit from the cilium remains largely unknown. Ptch1 can be ubiquitinated by the E3 ligases Smurf1 and Smurf2, followed by lysosomal degradation[16]. However, this mechanism may not apply to Ptch1 exit from the cilium[17]. Another recent study showed that Ptch1 in commissural neuron growth cones is internalized via Numb, an adaptor for clathrin-mediated endocytosis.

[1]Department of Molecular and Cell Biology, University of California, Merced, Merced, CA 95340, USA. [2]Montreal Clinical Research Institute (IRCM), Montreal, QC H2W 1R7, Canada. [3]Department of Anatomy and Cell Biology, McGill University, Montreal, QC H3A 0G4, Canada. [4]Department of Biology, McGill University, Montreal, QC H3A 0G4, Canada. [5]Departments of Genetics, of Biology, and by courtesy, of Chemistry, Stanford University, Stanford, CA, USA. [6]Chan Zuckerberg Biohub, San Francisco, CA, USA. [7]Department of Medicine, University of Montreal, Montreal, QC H3T 1J4, Canada. [8]Department of Biochemistry, University of Washington, Seattle, WA 98195, USA. [9]Present address: Interline Therapeutics, South San Francisco, CA, USA. [10]These authors contributed equally: Xiaoliang Liu, Patricia T. Yam. ✉e-mail: frederic.charron@ircm.qc.ca; xge2@ucmerced.edu

This process is required for non-canonical Hh signaling and growth cone turning up Shh gradients[18]. Whether Numb acts as an endocytic adaptor to mediate Ptch1 departure from the cilium in canonical Hh signaling remains to be determined.

One approach to understanding signaling mechanisms in the cilium is to catalog the protein components in the cilium over the course of signal transduction. This has been technically challenging due to the miniature size of the cilium, which accounts for about 1/10,000 of the cytoplasmic volume. Proximity-based labeling tools provide a powerful method to specifically label and identify ciliary proteins. APEX and its variant APEX2, engineered peroxidases, have been used to specifically label proteins in the cilium[19–21]. Intriguingly, when the enzyme was targeted to different sub-ciliary domains by fusing to distinct "tagging" proteins, unique cohorts of cilium proteins were discovered.

In this study, we leveraged a mechanistically different proximity labeling tool, TurboID, to plot ciliary proteins over the course of Hh signal transduction. TurboID is an engineered promiscuous biotin ligase that labels neighboring proteins and avoids the need for toxic labelling conditions dependent on hydrogen peroxide[22]. Using a truncated form of the membrane-associated ciliary protein Arl13b to tag TurboID to the cilium, our ciliary proteome recovered a wide array of ciliary protein candidates. Among the new candidates, we investigated Numb, an endocytic adaptor involved in signaling regulation and cell fate specification[23–26]. We found that Numb localizes to the ciliary pocket, a cytoplasmic membrane region at the base of the cilium that "folds back" to ensheath the cilium. Moreover, we found that Numb interacts with Ptch1 and recruits Ptch1 to clathrin-coated vesicles (CCVs), thereby promoting Ptch1 exit from the cilium. Numb loss in NIH3T3 cells reduces the plateau levels of Hh signaling. Furthermore, in mouse embryonic stem cell (ESC)-derived spinal cord neural progenitors (NPCs), Numb depletion blocks cell differentiation into an identity that requires high Hh activity. Genetic ablation of Numb in the developing cerebellum decreased the proliferation of GCPs, a Hh-signaling dependent process, resulting in reduced cerebellar size. Overall, our study highlights a profile of membrane-associated proteins in the cilium, and demonstrates the role of Numb as a regulator of endocytosis in the ciliary pocket, a process crucial for the transduction of cilium-mediated cell signaling.

## Results

### Cilium-TurboID selectively biotinylates ciliary proteins

Many signaling proteins are membrane proteins or recruited to the membrane during signal transduction. To label these proteins in the primary cilium, we directed TurboID to the juxtamembrane region of the cilium. We chose Arl13b to fuse with TurboID since Arl13b is anchored to the ciliary membrane via lipid modification[27,28], maximizing the chance of protein biotinylation in the juxtamembrane region. As reported before, overexpressing full-length Arl13b leads to markedly elongated cilia in NIH3T3 cells[29] (Supplementary Fig. 1a). To determine which domains of Arl13b could retain ciliary localization without altering cilium length, we generated a series of truncated Arl13b constructs and transiently expressed them in NIH3T3 cells (Fig. 1a and Supplementary Fig. 1a). We found that the N + RVEP + PR construct fits this requirement. This construct contains the N-terminal amphipathic helix (containing a palmitoylation modification for membrane association), the previously reported cilium targeting sequence RVEP, and the Proline-Rich domain at the C-terminus[30] (Fig. 1a). In contrast, the construct containing only the PR domain showed no ciliary localization (Supplementary Fig. 1a, b).

We next fused TurboID to N + RVEP + PR (cilium-TurboID), and to the PR domain (non-cilium-TurboID) as a control for cytosolic proteins. We then generated stable NIH3T3 cell lines that express cilium- or non-cilium-TurboID via an integral lentiviral system. Confocal microscopy confirmed that cilium-TurboID is exclusively localized to

the primary cilium, whereas non-cilium-TurboID is diffusely distributed in the cytoplasm (Fig. 1b). Furthermore, 10 min labeling with biotin selectively labeled proteins in the cilium in cilium-TurboID cells, whereas cytosolic proteins were labeled in non-cilium-TurboID cells (Fig. 1b).

To ensure that the transgenes did not impair cilium function, we measured cilium length and Hh signaling activity in the stable cell lines. We found that the cilium lengths were comparable between WT (parental NIH3T3 cells), cilium-TurboID and non-cilium-TurboID cells (Supplementary Fig. 2a). We then determined the Hh signaling activity by measuring the transcript levels of *Gli1*, a Hh target gene. We found that Shh-induced *Gli1* levels were comparable between the WT and stable cell lines (Supplementary Fig. 2b). Further, Shh-induced cilium transport of Smo and Gli2 were indistinguishable between WT and the stable cell lines (Supplementary Fig. 2c–f). Therefore, cilium morphology and Hh signaling are normal in these stable cell lines. Thus, we developed a method to exclusively target TurboID to the juxtamembrane microdomain in the primary cilium without impacting cilium functions.

### Quantitative proteomics with cilium-TurboID highlights membrane-associated ciliary proteins

We then employed the stable cell lines to label and isolate endogenous ciliary proteins. We labeled cells with biotin for 10 min and captured the biotinylated proteins with streptavidin beads[22]. Proteins were analyzed by Western blotting (Fig. 2a). Streptavidin-HRP detection of the biotinylated proteins showed that biotin labeling drastically increased the overall levels of biotinylated proteins captured by the streptavidin beads. We used PDGFRα, a receptor known to localize to the cilium, as a positive control. PDGFRα was pulled down from cilium-TurboID cells, but not non-cilium-TurboID cells (Fig. 2a). These results confirm that cilium-TurboID is effective in labeling proteins in the cilium.

Next, we designed a quantitative proteomic experiment to capture the dynamic protein transport in the cilium in response to Shh stimulation. To control for background biotinylation, we included conditions that omitted either Shh stimulation or biotin labeling. To control for non-ciliary protein biotinylation by TurboID, we collected samples from stimulated and labeled non-cilium-TurboID cells (Fig. 2b). We prepared each condition with three independent replicates. After on-beads digestion, each sample was labeled with one channel of the 16plex tandem mass tag (TMT) labeling kit. A reference channel was included in which equal volumes from each of the 12 samples were pooled. After that, the 13 samples were multiplexed and analyzed by liquid chromatography (UPLC)/MS-MS (Fig. 2b; channel distribution in Supplementary Fig. 3a).

For data analysis, we defined the relative protein abundance as the ratio of normalized abundance in each channel over the reference channel. Only proteins with more than 7 unique peptides were used in data analysis. We determined candidate ciliary proteins via statistical analyses of the relative enrichment between the cilium-TurboID and the non-cilium-TurboID datasets. To be scored as ciliary proteins in the absence of Shh stimulation, candidates had to fulfill four criteria: (1) greater than 2-fold enrichment (TMT ratio > 2) in the labeled cilium-TurboID samples over the non-labeled cilium-TurboID samples, (2) and over the labeled non-cilium-TurboID samples; (3) statistically significant ($p$ value < 0.05) enrichment in the labeled cilium-TurboID samples versus the non-labeled cilium-TurboID samples, and (4) versus the labeled non-cilium-TurboID samples (Supplementary Fig. 3b, c). 788 proteins meet these criteria (Supplementary Data 1). To be scored as ciliary proteins after Shh stimulation, candidates had to fulfill the same four criteria, comparing between the labeled Shh-stimulated cilium-TurboID samples and the non-labeled Shh-stimulated cilium-TurboID samples, and the labeled Shh-stimulated non-cilium-TurboID samples (Supplementary Fig. 3d, e). 574 proteins meet these criteria

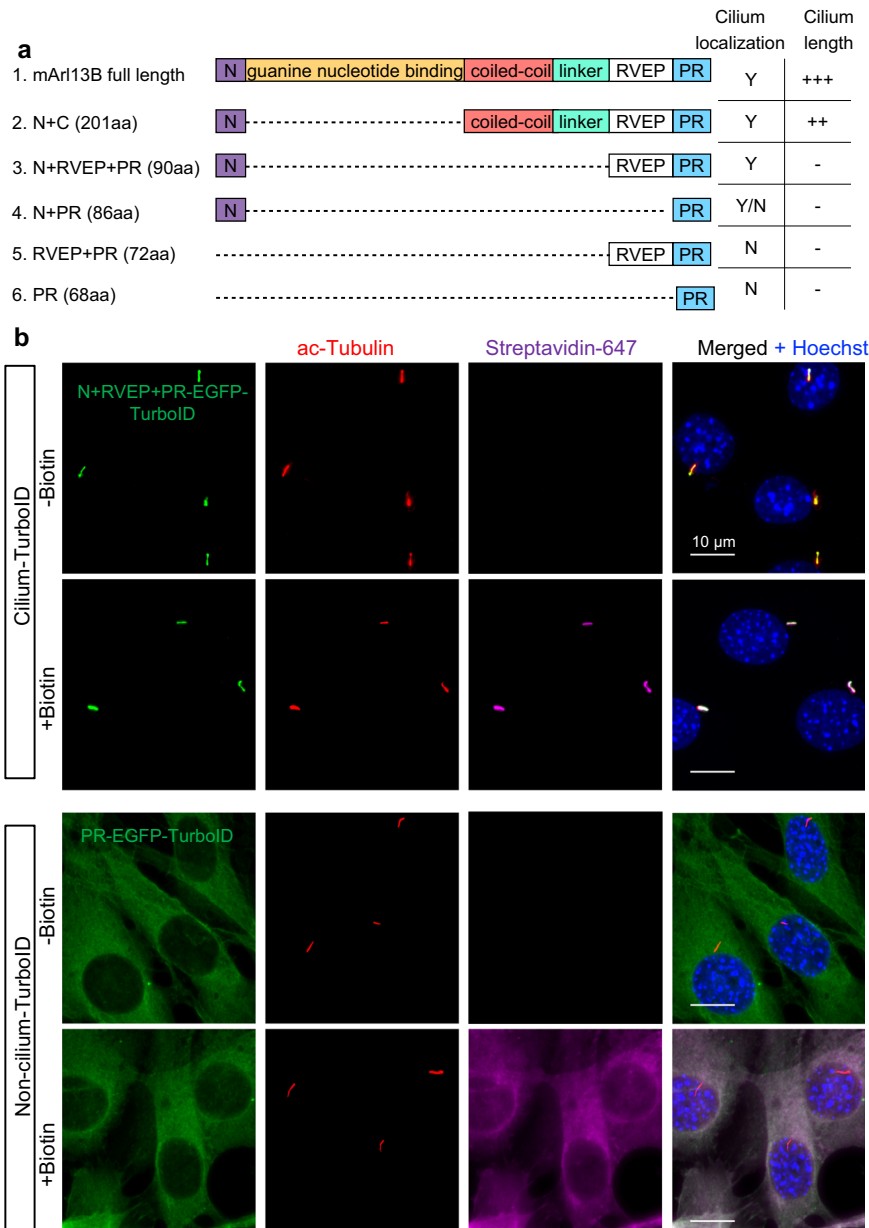

**Fig. 1 | Cilium-TurboID selectively biotinylates ciliary proteins. a** Schematics of the truncated Arl13b constructs used. Y and N indicate whether the corresponding construct localizes to the cilium or not, respectively. Y/N: this construct localizes to both the cilium and the cytosol. "+": the degree of cilium elongation in NIH3T3 cells transiently transfected with the corresponding constructs. "-": cilium length is not impacted. **b** Immunofluorescence of stable NIH3T3 cell lines expressing cilium-TurboID (N + RVEP-PR-EGFP-TurboID) or non-cilium-TurboID (PR-EGFP-TurboID) before and after biotin labeling. Cells were serum-starved for 24 h, incubated with 50 μM biotin for 10 min at 37 °C, and fixed for immunostaining. The cilium is labeled by immunostaining for acetylated-tubulin (red, ac-tubulin), and the biotinylated proteins were detected via streptavidin-Alexa Fluor 647 (magenta). Cilium-TurboID and non-cilium TurboID were detected by immunostaining for EGFP (green). Scale bars, 10 μm in all panels. Experiments were performed three times with similar results.

(Supplementary Data 1). Finally, we took the union of the above two categories and defined these proteins as ciliary candidates regardless of Shh stimulation. We obtained 800 such ciliary candidates (Supplementary Data 2). Correlation of biological replicates and a heatmap of each protein's relative abundance demonstrated a high reproducibility across the triplicates (Fig. 2c; Supplementary Fig. 4a–d).

We then compared the intensity of the above identified ciliary candidates between the Shh-stimulated and unstimulated conditions (Fig. 2d, Supplementary Data 2). 30 proteins showed statistically significant changes (TMT ratio > 1.2 or <0.83, $p < 0.05$). Among them is the Hh transducer Smo. It is important to note that a lack of statistical significance does not imply that the candidates do not

respond to Hh signaling, owing to a variety of reasons such as (1) the low abundance of the protein in the cilium, and (2) preferential biotinylation of proteins in juxtamembrane subdomain in our experimental settings. Indeed, a few proteins known to translocate to the cilium in response to Hh activation show larger $p$ value, such as Gli2 and PALD1[20].

Among the total ciliary candidates, 108 have been reported before to localize to the cilium (Supplementary Fig. 4e). Analysis of the ciliary candidates with enrichment of Gene Ontology (GO) terms in Metascape shows that the largest cohort are membrane-associated proteins (Fig. 2e). These results validated our strategy of targeting TurboID to the ciliary juxtamembrane domain.

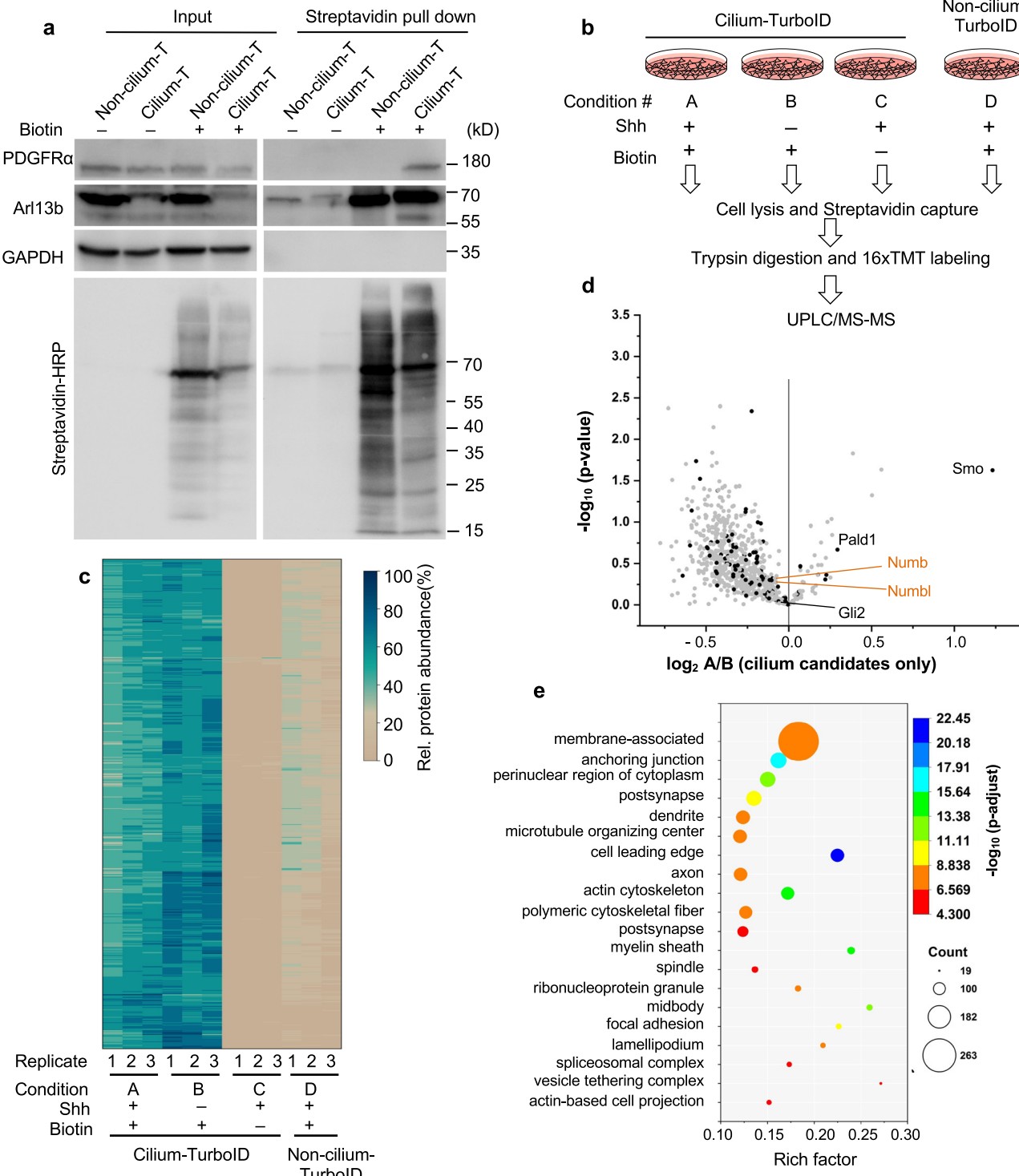

## Numb localizes to the ciliary pocket and clathrin-coated vesicles (CCVs)

Among the newly identified ciliary candidates, Numb is particularly interesting. *Drosophila* Numb has important roles in asymmetric cell division[23], and mammalian Numb regulates neurogenesis in the developing brain[24–26]. Our proteomic results revealed Numb as a potential ciliary protein. The mass spectrometry intensities of Numb in the cilium display a reduced tendency following Shh stimulation, although no statistically significant changes are observed (Fig. 2d). To validate its ciliary localization, we expressed V5 tagged Numb in NIH3T3 cells. We found that in addition to its reported distribution in

the cytosol as puncta[31,32], Numb-V5 also overlaps with the ciliary marker Arl13b in 33% of transfected cells (Fig. 3a, Supplementary Fig. 5a, b). However, the Numb immunofluorescence signal is not distributed along the entire cilium; rather, it is restricted to the lower section of the cilium.

The localization pattern of Numb in the cilium does not resemble any of the known cilium subdomains, such as the transition zone[33], inversin zone[34], or Evc zone[35]. Rather, it resembles the recently characterized ciliary pocket, a cytoplasmic membrane region at the bottom of the cilium that often folds back to ensheath the ciliary shaft[36,37]. To characterize Numb's localization in the ciliary pocket, we co-

**Fig. 2 | Quantitative proteomics with cilium-TurboID identified novel ciliary proteins. a** Biotinylated proteins from cell lysates of cilium-TurboID and non-cilium-TurboID cells were isolated by streptavidin beads and analyzed by Western blotting. The overall biotinylated proteins were detected by streptavidin-HRP. PDGFRα is a receptor known to localize to the cilium. In the Arl13b blot, the bands at 70kD represent the transgene, and the bands at 55 kD represent endogenous Arl13b. Experiments were performed three times with similar results. **b** Design and workflow of the cilium proteomics experiment. Samples were prepared in triplicate. After TMT labeling, all samples were pooled together before being fractionated by UPLC and analyzed by mass spectrometry. **c** Heatmap of the ciliary abundance of the 800 ciliary candidates. Clustering of the relative abundances of each identified protein (rows) in individual samples (columns) was performed based on Ward's minimum variance method. The relative abundance was calculated within one set of samples that contains all four conditions. For example, A1,

B1, C1, D1 belong to one set, and the relative abundance is calculated as A1/(A1 + B1 + C1 + D1) etc. The panel on the right side indicates the color scheme of relative abundances (percentage). **d** Volcano plot of statistical significance versus the average ratio of protein enrichment for all ciliary candidates. The ratio was calculated as TMT signal intensity of Shh-treated samples (A) compared to non-Shh treated samples (B) in the presence of biotin labeling. The analytical procedure for identifying ciliary candidates from the mass spectrometry results is described in supplementary Fig. 3. The previously reported ciliary proteins are highlighted as black dots. **e** GO enrichment analysis of cellular components was plotted according to the rich factor in Metascape. The top 20 enriched cellular components are represented in the scatter plot. The $p$ values in (**d**) and (**e**) were adjusted for multiple testing via the Benjamini-Hochberg procedure. Source data are provided as a Source Data file.

transfected NIH3T3 cells with Numb-mCherry and EHD1-GFP, a protein known to localize to the ciliary pocket[38](Fig. 3b). We then performed expansion microscopy, a procedure that physically expands the samples in an isotropic fashion to allow nanoscale resolution[39,40]. We found that Numb overlaps with EHD1 at the lower part of the cilium in the region flanking the cilium axoneme (Fig. 3b), suggesting that Numb localizes to the ciliary pocket. To further confirm this result, we performed immuno-electron microscopy in NIH3T3 cells expressing HA-tagged Numb. Notably, extensive electron-dense staining is observed in the ciliary pocket membrane (Fig. 3c). Thus, Numb localizes to the ciliary pocket.

In addition to the ciliary pocket, Numb immunofluorescence appeared as discrete puncta in the cytosol (white triangles in Fig. 3b). Since Numb is involved in clathrin-mediated endocytosis[32], these puncta are likely clathrin-coated vesicles (CCVs). To test this, we co-transfected NIH3T3 cells with Numb-GFP and clathrin-mCherry, and imaged cells via expansion microscopy. At the focal plane focusing on the bottom of the cilium, we found that at the region adjacent to the cilium, all Numb-containing puncta were clathrin positive, suggesting that these puncta are CCVs (Fig. 3d). Further, multiple double positive puncta are still in close proximity to the cilium (Fig. 3d, cyan triangles), indicating that these puncta are very likely CCVs emerging from the ciliary pocket[37]. Taken together, Numb localizes to the ciliary pocket and to CCVs derived from the ciliary pocket (Fig. 3e).

### Numb interacts with Ptch1 and incorporates Ptch1 into CCVs
The ciliary pocket is believed to play a crucial role in the endocytic control of ciliary proteins[36,37,41]. Our findings indicate that Numb is not only present in the ciliary pocket but also associated with CCVs linked to the ciliary pocket. Given Numb's established function as an adaptor that connects endocytosed cargoes with adaptins in the clathrin-mediated endocytic machinery[42,43], it is plausible that Numb may participate in endocytosis at the ciliary pocket. During Hh signal transduction, Ptch1 exit from the cilium is a critical step to ensure the maximal activation of Hh signaling. We hypothesized that Numb recruits Ptch1 to CCVs at the ciliary pocket, thereby mediating Ptch1 exit from the cilium and activating Hh signaling.

To test our hypothesis, we examined whether Ptch1 and Numb can interact. In 293 T cells, Ptch1-GFP co-immunoprecipitated with Numb-HA (Fig. 3f). We then tested whether endogenous Ptch1 and Numb form complexes with each other in ciliated cells. We immunoprecipitated endogenous Numb from lysates of NIH3T3 cells and found that endogenous Ptch1 co-immunoprecipitated with Numb (Fig. 3g). Therefore, Numb and Ptch1 form complexes with each other. Using Numb truncated proteins in co-immunoprecipitation assays, we found that deletion of either the N-terminus (aa1-25) or the PTB domain (aa26-173) abolished the interaction with Ptch1. Meanwhile, the N + PTB domain (aa1-173) is sufficient for interaction with Ptch1 (Supplementary Fig. 5a, c). These results indicate that Numb interacts with Ptch1 via its N-terminal and PTB domain.

Next, we determined whether Ptch1 is incorporated into Numb-containing CCVs. We co-expressed Numb and Ptch1 in NIH3T3 cells via lentiviruses that allow low expression levels to recapitulate the behavior of endogenous proteins. In the absence of Shh, only a limited number of Numb-positive puncta colocalize with Ptch1-YFP. Shh stimulation significantly increases the colocalization of Numb and Ptch1 within the puncta in the region surrounding the cilium (Fig. 3h, i). As a control, we co-expressed YFP with Numb-V5, and did not observe any significant overlap between Numb and YFP (Supplementary Fig. 6a). These results suggest that Shh stimulation promotes Ptch1 incorporation into Numb-containing CCVs.

### Numb is required for Shh-induced Ptch1 exit from the cilium
Since Shh increases the association of Numb with Ptch1 in CCVs, we hypothesized that Numb is required for Shh-induced Ptch1 exit from the cilium. To test this, we generated *Numb* knockout cells via CRISPR/Cas9 in NIH3T3 cells. We used two guide RNAs to target the first and third exons of *Numb* (Supplementary Fig. 7a), and isolated colonies of *Numb* knockout (KO) cells. We then validated biallelic indel mutations that eliminated Numb expression in two of the cell clones (Supplementary Fig. 7b, c). In these two clones, Numb protein was undetectable in Western blotting (Fig. 4a). We used these two cell clones in subsequent experiments.

Given that Ptch1 in the cilium in unstimulated cells is below the detection threshold of immunostaining[4], we used overnight SAG treatment to induce Ptch1 expression and localization to the cilium (Fig. 4b). We then determined the effect of Numb loss on Ptch1 exit from the cilium after Shh stimulation. In WT cells, Ptch1 intensity significantly decreased in the cilium after Shh stimulation. In contrast, in *Numb* KO cells, the ciliary Ptch1 intensity was unchanged after Shh stimulation (Fig. 4b, c). In another experiment, we expressed low levels of Ptch1-YFP via lentivirus in WT or *Numb* KO cells. Shh effectively induced Ptch1-YFP exit from the cilia in WT cells but not in *Numb* KO cells (Supplementary Fig. 8a, b). To exclude the potential impact of Numb loss on the formation of the ciliary pocket, we visualized the pocket after expansion microscopy. The morphology of the ciliary pocket, marked by EHD1-GFP, showed no discernible differences between the WT and *Numb* KO cells (Supplementary Fig. 8c). In conclusion, Numb loss hinders Ptch1 exit from the cilium.

### Numb loss diminishes the plateau levels of Hh signaling
In the absence of Hh stimulation, Ptch1 in the cilium suppresses Smo activation (Fig. 5a). Hh stimulation induces Ptch1 exit from the cilium, which is critical for the activation of Hh signaling. We predicted that without Numb, Hh signaling would be suppressed due to impaired Ptch1 exit from the cilium. To test this, we evaluated Hh signaling in *Numb* CRISPR/Cas9 KO cells. We stimulated cells with Shh and assessed Hh signaling activity by measuring the transcript levels of the Hh target genes, *Gli1* and *Ptch1*. As predicted, Numb loss severely attenuated Hh signaling activity, particularly at the higher doses of Shh.

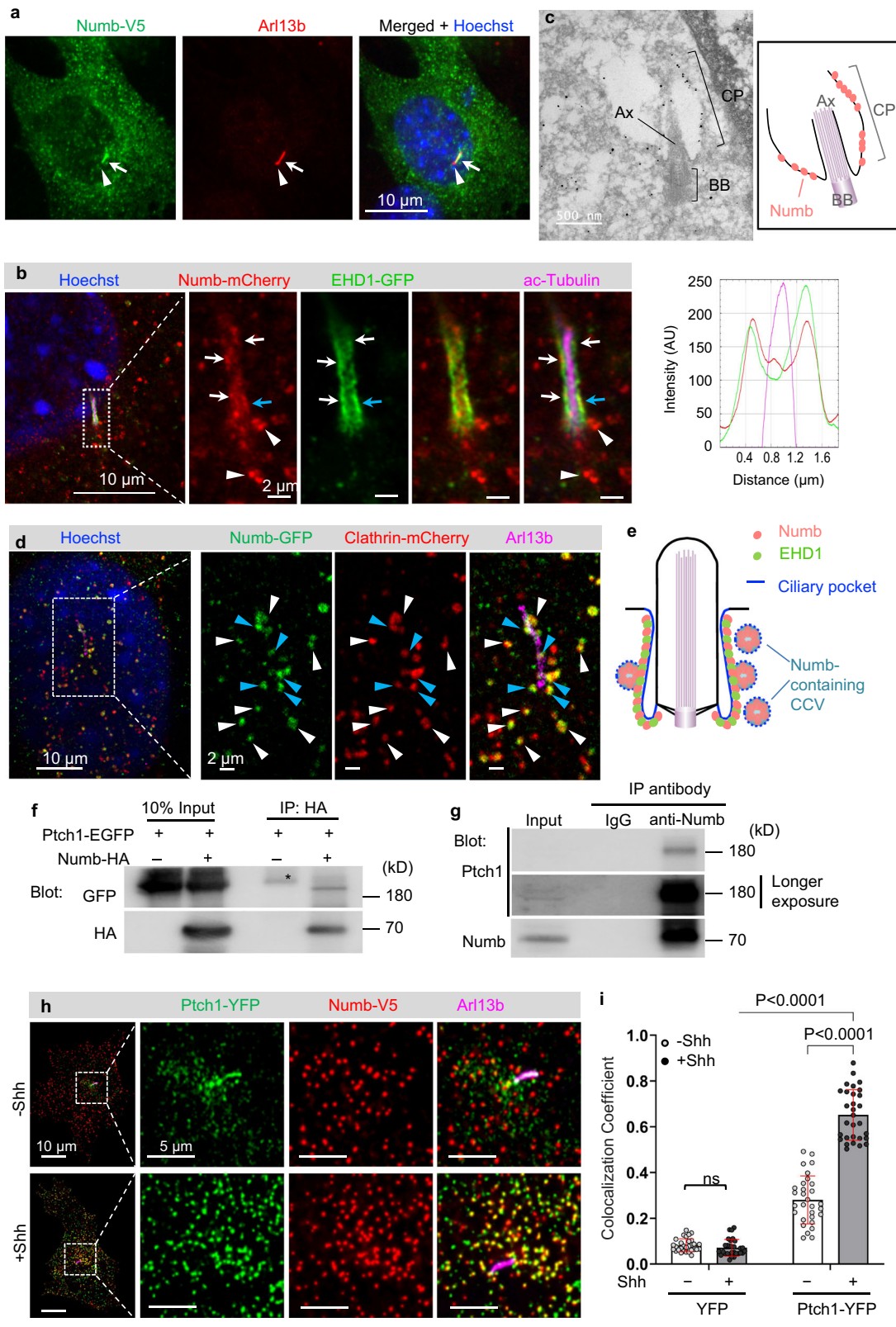

The lower level of Hh signaling was modestly impacted, while the plateau levels of Hh signaling were significantly diminished in *Numb* KO cells (Fig. 5b, c). To exclude potential off-target effects of CRISPR/Cas9, we infected the *Numb* KO cells with lentiviruses that express Numb-V5. Restoring Numb expression in *Numb* KO cells reinstates Shh-induced Hh signaling to levels comparable to those in WT cells (Fig. 5d).

To confirm the effect of Numb loss on Hh signaling, we silenced Numb expression with an alternative method. We infected cells with lentiviruses that express *Numb* shRNA and stimulated cells with the highest dose of Shh. Two independent shRNA constructs against *Numb* attenuated Shh-induced *Gli1* and *Ptch1* transcription to the same extent as *Numb* CRISPR/Cas9 mediated KO (Supplementary Fig. 9a–c). Further, the attenuated Hh signaling was rescued by the expression of the full-

**Fig. 3 | Numb localizes to the ciliary pocket and incorporates Ptch1 into clathrin-coated vesicles. a** Numb localizes to the lower section of the primary cilium when expressed in NIH3T3 cells. Cilium is highlighted by Arl13b staining. White arrow points to the lower section of the cilium; white triangle points to the ciliary tip where Numb immunofluorescence is absent. **b** NIH3T3 cells were co-transfected with Numb-mCherry and EHD1-GFP, and subjected to expansion microscopy. The displayed image corresponds to a single focal plane encompassing the lower segment of the cilium axoneme. Arrows point to the ciliary pocket where Numb and EHD1 exhibit colocalization; white triangles indicate Numb-containing puncta located adjacent to the cilium. The right panel shows the results of the linescan at the position marked by the cyan arrow. **c** Representative image of ImmunoEM on NIH3T3 cells expressing Numb-HA. CP Ciliary pocket, Ax axoneme, BB basal body. **d** NIH3T3 cells were co-transfected with Numb-GFP and clathrin-mCherry, and subjected to expansion microscopy. The displayed image

corresponds to a single focal plane. White triangles point to the Numb-containing CCVs. Cyan triangles point to Numb-containing CCVs still in conjunction with the ciliary pocket. **e** Diagram illustrating Numb's localization to the ciliary pocket and to CCVs situated in proximity to the ciliary pocket. **f** Co-immunoprecipitation results from 293 T cells transfected with Ptch1-EGFP and Numb-HA. Asterisk indicates a non-specific band. **g** Co-immunoprecipitation results from NIH3T3 cells treated with Shh. **h** Representative images showing that Ptch1 is incorporated into Numb-containing CCVs near the ciliary pocket. **i** Colocalization coefficient of Numb-V5 and Ptch1-YFP or YFP. A total of 30 cells were quantified for each experimental condition. All experiments in (**a**–**h**) were performed three times with similar results. Data in (**i**) are presented as mean ± SD. Statistics: Two-way ANOVA with multiple comparisons (Tukey test), ns not significant. Source data are provided as a Source Data file.

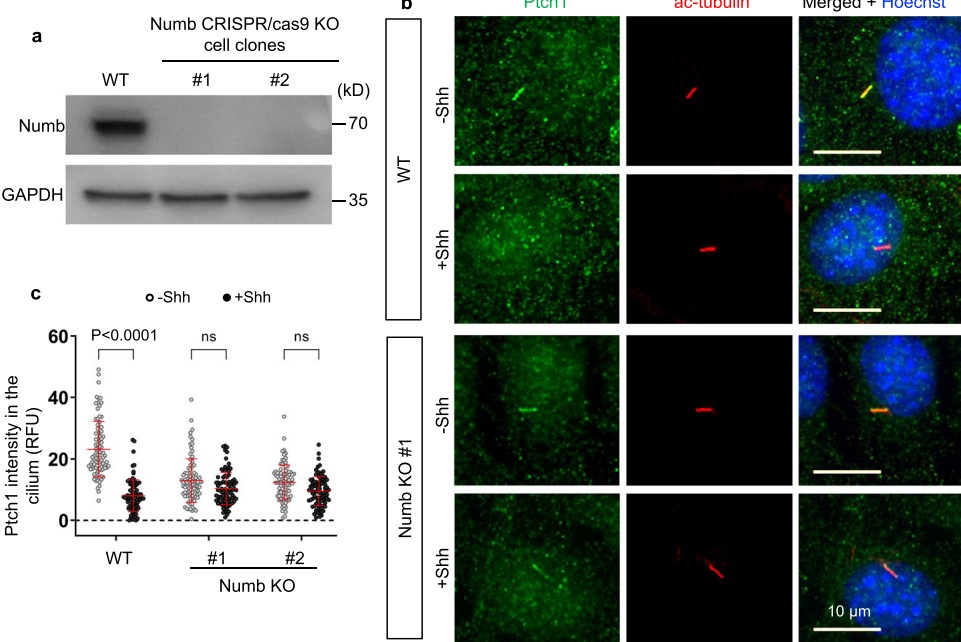

**Fig. 4 | Numb is required for Shh-induced Ptch1 exit from the cilium. a** Western blot analysis showing the absence of Numb protein in the *Numb* CRISPR/Cas9 knockout cell clones. Experiments were performed three times with similar results. **b** Wild type (WT) or *Numb* KO cells were pre-treated with SAG for 24 h to induce Ptch1 expression and cilium accumulation, followed by Shh stimulation for 1 h, and

subsequently fixed for immunofluorescence staining. The cilia were marked by acetylated-tubulin (red). **c** Quantification of Ptch1 intensity in the cilium. Data are shown as mean ± SD. A total of 87 cells were quantified for each experimental condition. Statistics: Two-way ANOVA with multiple comparisons (Tukey test). ns not significant. Source data are provided as a Source Data file.

length Numb, but not any of the truncated Numb variants that lack either the Ptch1-binding domain (N-terminus and PTB) or the C-terminal domains known to bind to adaptins, components of clathrin-mediated endocytic machinery[42,43] (Supplementary Fig. 9d). Thus, both the Ptch1-binding domain and the adaptin-interacting domains are essential for Numb's positive role in Hh signal transduction.

Numb has been reported to regulate Gli1 protein degradation[44]. We hence tested the impact of Numb loss on the protein levels of Gli1 with Western blot analysis. We found that Gli1 protein levels are significantly reduced in *Numb* KO cells compared to WT cells (Fig. 5e, f). The reduced Gli1 protein levels are most likely a consequence of attenuated Gli1 transcription.

Given that Numb exerts its regulatory effect at the level of Ptch1, activating the pathway downstream of Ptch1 should turn on Hh signaling in *Numb* KO cells (Supplementary Fig. 10a–d). To test it, we first stimulated cells with SAG, a Smo agonist. SAG induced Hh signaling in *Numb* KO cells to levels comparable to those in WT cells (Fig. 5g). Second, we transfected cells with SmoM2, a constitutively active Smo mutant that triggers Hh signaling independent of Shh[45]. SmoM2 triggered Hh signaling in *Numb* KO cells to levels comparable to WT cells

(Fig. 5h). Third, we knocked-down Sufu, a core component of Hh signaling that suppresses the pathway at the level of Gli. Sufu knockdown turned on Hh signaling independently of Shh in *Numb* KO cells (Supplementary Fig. 10e, f). Finally, we specifically blocked the ciliary activity of PKA, the kinase that exerts its inhibitory roles on Hh signaling in cilia[21,46,47]. Expressing a cilium-targeting PKI (a PKA peptide inhibitor tagged to Arl13b-N-RVEP-PR) triggered Hh signaling to a similar magnitude in *Numb* KO cells and WT cells (Supplementary Fig. 10g). In summary, after Numb depletion, activating Hh signaling downstream of Ptch1 triggers the full activation of Hh signaling, circumventing the inhibitory effects caused by Numb loss.

## Numb loss blocks the activation of Gli transcription factors
We next sought to determine how Numb loss impacts Hh signaling events downstream of Ptch1. We first examined the cilium accumulation of Smo. We found that following Shh stimulation, the intensity of Smo in the cilium of *Numb* KO cells was comparable to that in WT cells (Supplementary Fig. 11a, b). It is known that Smo may accumulate in the cilium in an inactive conformation[48]. We hence examined the Gli transcription factors downstream of Smo.

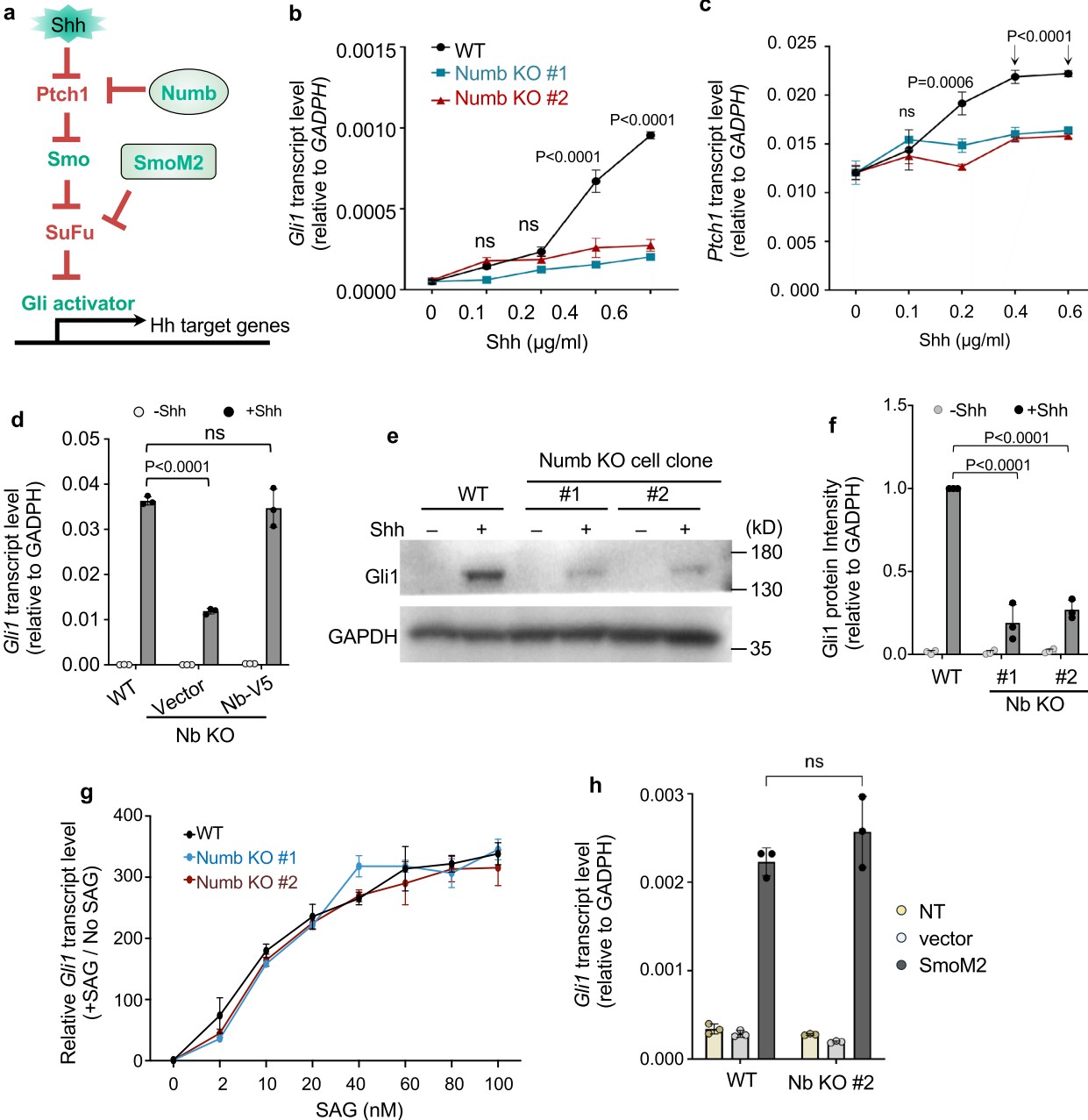

**Fig. 5 | Numb is required for maximal activation of Hh signaling. a** Diagram of the Hh signaling cascade. Numb regulates Hh signaling at the level of Ptch1; SmoM2 actives Hh signaling independent of Shh. **b**, **c** Hh signaling levels in WT or *Numb* KO cells were assessed by the transcript levels of the Hh-target genes, *Gli1* and *Ptch1*, via qPCR. **d** Hh signaling levels in WT cells and *Numb* KO cells that express Numb-V5 or the blank vector. Hh activity was assessed by *Gli1* transcript levels. **e**, **f** Western blot analysis and quantification of Gli1 protein levels in WT and *Numb* KO cells. Prior to harvesting, cells were stimulated with 1 µg/ml Shh in the low

serum culture medium for 24 h. **g** SAG-induced Hh signaling levels in WT cells and *Numb* KO cells. Prior to harvesting, cells were treated with the indicated doses of SAG in the low serum culture medium for 24 h. **h** Hh signaling activity in WT cells and *Numb* KO cells infected with lentiviruses that express SmoM2. All experiments were repeated three to four times with similar results. Data are shown as mean ± SD. Statistics in (**b**, **c**, **d**, **f**, **g**, **h**): Two-way ANOVA with multiple comparisons (Tukey test). ns not significant. Source data are provided as a Source Data file.

At the resting state, the transcription of Hh target genes is suppressed by Gli3R. Low levels of Shh stimulation terminate the production of Gli3R, lifting the inhibition on Hh signaling. Conversely, high levels of Shh stimulation activate Gli2, resulting in maximal activation of Hh signaling. We examined the cilium translocation of Gli2, a critical step for Gli2 activation. We found that Shh-induced cilium transport of Gli2 was markedly reduced in *Numb* KO cells compared with WT cells (Fig. 6a, b). We then analyzed the proteolysis of Gli3, a process responsible for generating Gli3R. The results show that Gli3R production was ceased by Shh stimulation in both WT and *Numb* KO

cells (Fig. 6c, d). These results suggest that without Numb, cells retain the capacity to block the generation of the repressive factor Gli3R, but have deficiencies in activating Gli2.

The homolog Numblike (Numbl) functions redundantly with Numb under specific circumstances such as neurogenesis in the cerebral cortex[24–26]. We therefore analyzed the localization of Numbl and its role in Hh signaling. We found that Numbl-V5 also localizes to the lower section of the primary cilium. However, unlike Numb, Numbl does not appear as discrete puncta in the cytosol (Supplementary Fig. 12a). This distinction indicates that Numbl may not be involved in

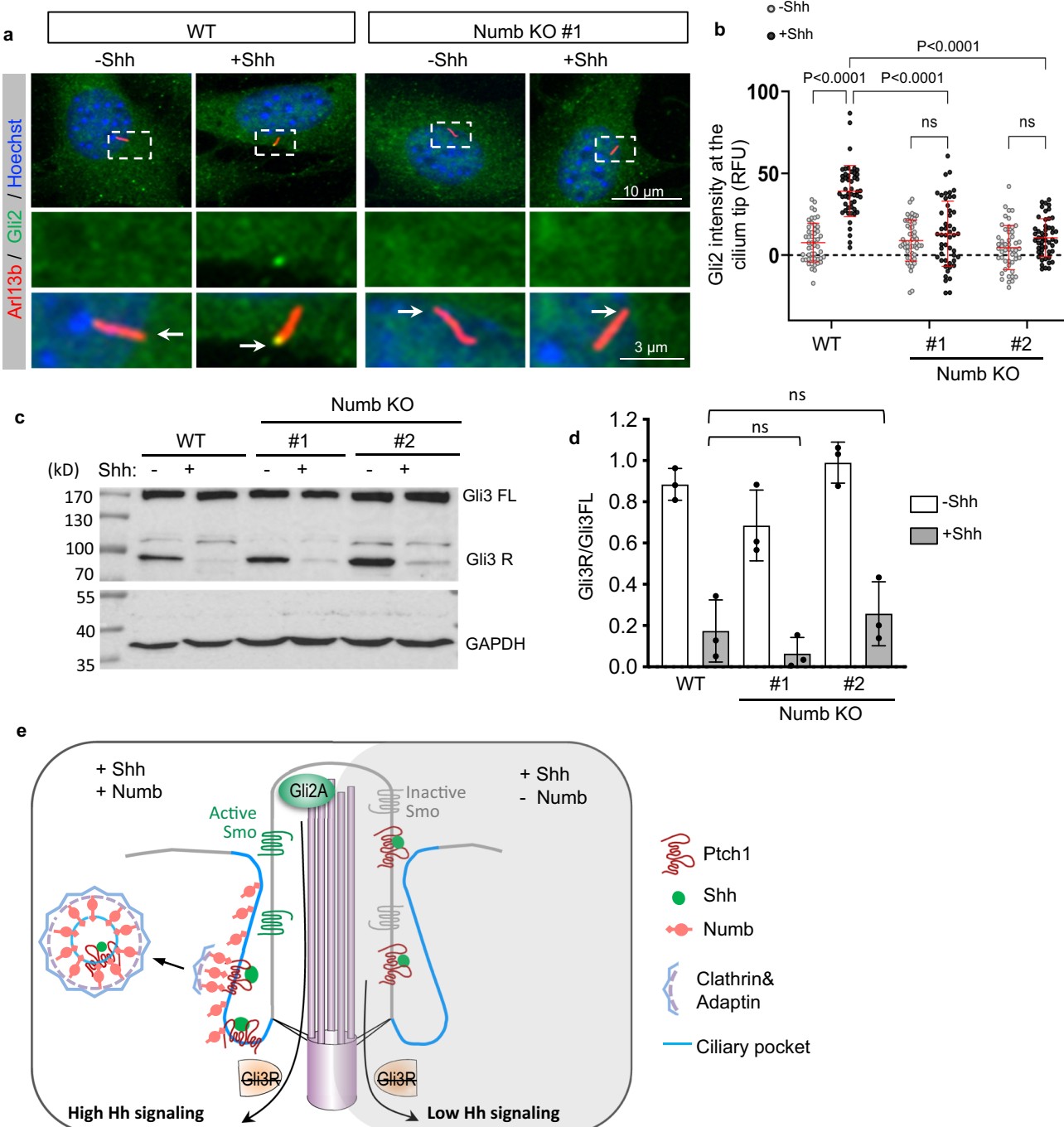

**Fig. 6 | Numb loss blocks the activation of Gli transcription activator.**
**a** Immunofluorescence staining of endogenous Gli2 in the cilia of WT or *Numb* KO cells. The cells were treated in low serum medium for 24 h, with or without Shh. Enlarged views of highlighted areas are displayed at the bottom. Arrows point to Gli2 fluorescence signal at the cilium tips. **b** Quantification of Gli2 fluorescence intensity at the cilium tips. A total of 50 cilia were quantified per condition. RFU, relative fluorescence unit. **c, d** Western blot results and quantification of Gli3 processing in WT and *Numb* KO cells. Cells were stimulated with Shh for 24 h prior to harvest. **e** Schematic of Numb's role during the activation of Hh signaling. Upon Shh binding to Ptch1, Numb in the ciliary pocket (blue lines) recruits Ptch1-Shh

complex into clathrin-coated vesicles, thereby facilitating the removal of Ptch1 from the cilium. Ptch1's efficient clearance from the cilium sets the stage for the complete activation of Smo and Gli2 transcription factors. This activation ultimately culminates in the maximal activation of Hh signaling. In the absence of Numb, Ptch1 remains in the cilium even after it binds to Shh. This leads to only a partial activation of Smo, which ceases Gli3R production but is insufficient to activate Gli2. As a result, Hh signaling is only moderately activated. Results from three independent experiments are analyzed. Data are shown as mean ± SD. Statistics in (**b, d**): Two-way ANOVA with multiple comparisons (Tukey test). ns, not significant. Source data are provided as a Source Data file.

endocytosis. We then determined whether Numbl is required for Hh signal transduction. We silenced *Numbl* with shRNA and found that it does not impact Shh-induced Hh signaling (Supplementary Fig. 12b, c). Further, silencing *Numbl* in *Numb* KO cells does not further reduce Shh-induced Hh signaling (Supplementary Fig. 12c). Thus, Numbl is

likely not involved in Ptch1 endocytosis from the ciliary pocket during Hh signal transduction.

Collectively, our results suggest a molecular model of Numb's roles at the ciliary pocket in Hh signal transduction (Fig. 6e). As a molecular adaptor for endocytosis, Numb connects the cargo Ptch1 to

components of the clathrin-mediated endocytic machinery. Specifically, the N + PTB domain of Numb binds to Ptch1, while its C-terminus interacts with adaptins, thereby incorporating Ptch1 into CCVs. This in turn facilitates Ptch1's departure from the cilium. While there is a basal level of CCV formation incorporating Ptch1, the binding of Shh to Ptch1 significantly increases the rate of CCV formation, accelerating Ptch1 exit from the cilium. This allows full activation of Smo and Gli2 to induce high levels of Hh signaling. In the absence of Numb, Ptch1 remains in the cilium even after it binds to Shh. The persistent presence of Ptch1 in the cilium impedes the full activation of the signaling steps downstream of Ptch1. Subsequently, cells can cease Gli3R generation but are unable to activate Gli2. As a result, Hh signaling is only moderately activated in the absence of Numb.

## Numb is required for high-level Hh responses in spinal cord neural progenitor cells (NPCs)

Hh signaling plays a crucial role in neural tube patterning, where a gradient of Shh ligand secreted by the notochord and floor plate determines distinct neural identities[49,50]. The identity of each cell fate corresponds to the expression of a specific set of transcription factors within distinct progenitor domains. Along the ventral-dorsal axis, the expression of transcription factors corresponds to the magnitude of Hh signaling. High Hh signaling level specifies *Nkx2.2* expression; medium-to-low signaling, *Olig2* and *Nkx6.1*; and no Hh signaling, *Pax6* (Fig. 7a). This morphogenic activity of Hh ligands can be recapitulated in cultured spinal cord neural progenitor cells (NPCs) derived from embryonic stem (ES) cells[51,52]. In cultured NPCs, varying levels of Hh signaling elicit the expression of distinct transcription factors, which can be assayed by qPCR or immunostaining.

To evaluate the role of Numb in NPC differentiation, we employed the same gRNA for NIH3T3 cells and knocked out *Numb* via CRISPR/Cas9 in mouse ES cells (Supplementary Fig. 13a). Since Numb primarily influences the plateau levels of Hh signaling and has minimal effect on the low levels, we exposed cells to high Shh concentrations during their differentiation. As in NIH3T3 cells, Numb loss in NPCs markedly reduced Shh-induced Hh signaling (Fig. 7b, c). Consequently, we observed a substantial reduction in the expression of *Nkx2.2*, in contrast to WT cells. There was moderate impact on the expression of *Nkx6.1*, and no significant effect on *Olig2* (Fig. 7d–f). These results suggest that Numb loss blocks the differentiation of NPCs into Nkx2.2-positive progenitors, a cell identity reliant on high Hh signaling. To confirm this result, we assayed the differentiation of NPCs into Nkx2.2-positive progenitors via immunostaining. Following high Shh induction, there is a substantial reduction in the percentage of *Numb* KO cells positive for Nkx2.2, compared to WT cells (Fig. 7g, h). In contrast, Numb loss had no significant impact on the differentiation of NPCs into Nkx6.1- or Olig2-positive cells (Supplementary Fig. 13b–d).

Collectively, the results in spinal cord NPCs substantiate our findings in NIH3T3 cells. In the developing spinal cord, along the ventral-dorsal axis, the Shh gradient is converted into opposing gradients of GliA (ventral high, dorsal low) and GliR (dorsal high, ventral low) (Fig. 7a, i). In Gli mutant mice the activation of the ventral marker Nkx2.2 relies on GliA, while the expression of more dorsal markers, Nkx6.1 and Olig2, is curtailed by GliR binding[50,53]. Likewise, in Numb-depleted cells, Gli2 activation is impeded (Fig. 6a), leading to suppression of NPC differentiation into Nkx2.2-positive cells. Meanwhile, Numb-depleted cells still eliminate Gli3R, and the remaining moderate levels of Hh signaling are sufficient to induce NPC differentiation into Nkx6.1- and Olig2-positive cells (Fig. 7i).

## Numb is required for Hh signaling and Shh-induced proliferation of GCPs

During cerebellum development, Shh acts as a mitogen to stimulate the proliferation of granule cell precursors (GCPs) in the external granule layer (EGL) from E17.5 to early postnatal development[54–57].

Previous studies have revealed the expression of Numb in cerebellar GCPs using RNA-seq[58], immunostaining[59], and in situ hybridization[60] (Supplementary Fig. 15a). We thus determined the subcellular localization of Numb in GCPs. We cultured GCPs obtained from the cerebellum of P7 mouse pups, and infected the GCPs with lentiviruses expressing either Numb-GFP or Numb-mCherry (Fig. 8a). Immunostaining revealed that both Numb constructs localize to the bottom section of the primary cilium, in addition to its punctate localization in the cytosol. This pattern resembles the localization of Numb observed in NIH3T3 cells (Fig. 3a).

Given the requirement for Numb in Hh signal transduction in NIH3T3 cells, we hypothesized that Numb may also be required for Hh signaling in GCPs. To test this, we first assessed the effect of Numb loss on Hh signaling in cultured GCPs. We infected primary cultured GCPs with lentiviruses expressing shRNA against *Numb*. Two independent shRNA effectively silenced *Numb* expression and significantly diminished Shh-induced *Gli1* transcription (Supplementary Fig. 14a, b). The cilium length in Numb knockdown GCPs remained comparable to that in control cells (Supplementary Fig. 14c). We then assessed Shh-induced Ptch1 exit from the cilium. While Shh markedly induced the clearance of Ptch1 from the cilia in control GCPs, this effect was blocked after Numb knockdown (Supplementary Fig. 14d, e). Further, Shh-induced Smo accumulation in the cilium is comparable in GCPs of Numb knockdown and control cells (Supplementary Fig. 14f, g). Hence, Numb positively regulates Hh signaling in GCPs, and exerts similar effect on Ptch1 and Smo as observed in NIH3T3 cells.

To further evaluate Numb's effect on Hh signaling in GCPs, we generated *Numb* conditional knockout (*Math1-cre⁺;Numbᶠ/ᶠ*) mice in the *Numbl*-null (*Numbl⁻/⁻*) background. We used *Math1-Cre* to mediate depletion of Numb from GCPs in the developing cerebellum. In primary cultured GCPs, Shh markedly induced Hh signaling in *Numbl*-null GCPs; however, removing *Numb* in a *Numbl*-null background (*Math1-cre⁺;Numbᶠ/ᶠ;Numbl⁻/⁻*) significantly reduced Hh signaling (Fig. 8b). Further, we found that SAG-induced Hh signaling levels were comparable in cells with or without *Numb* (Fig. 8c), corroborating our findings in NIH3T3 cells that Numb exerts its regulatory function at the level of Ptch1.

Next, we determined whether the reduced Hh signaling in GCPs after Numb depletion corresponds to a change in their proliferative response to Shh. We generated mice conditionally lacking *Numb* and *Numbl* (referred to as cDKO hereafter) in GCPs using the *Math1-Cre* driver. Western blot analysis on isolated GCPs showed effective ablation of Numb protein from the cDKO mice (Fig. 8f). Further, the cDKO cerebellum exhibits significantly lower Numb immunofluorescence intensity in the EGL (Fig. 8d, e), reflecting effective removal of Numb from GCPs. In primary cultured GCPs, stimulation with Shh for 48 h induced their proliferation in a dose-dependent manner, measured by ³H-thymidine incorporation. The GCPs from cDKO mice showed significantly lower proliferation in response to Shh (Fig. 8g). In contrast, cDKO GCPs showed a similar proliferation rate to control cells in response to the Smo agonist purmorphamine (Fig. 8g). These results align with our findings in NIH3T3 cells that Numb regulates Hh signaling upstream of Smo. We next examined if Numb loss reduced GCP proliferation in vivo in the cerebellum. We used the mitotic marker phospho-histone H3 (pH3) to measure the number of proliferating cells in the EGL. The results show that the EGL of cDKO cerebellum contains significantly fewer proliferating cells compared to the EGL of control mice (Fig. 8h, i). In summary, the reduced Hh signaling caused by Numb loss leads to reduced GCP proliferation in the developing cerebellum.

## *Numb;Numbl* cDKO mice show a decreased GCP population and reduced cerebellar size

The reduced GCP proliferation in the EGL may decrease the GCP population, which would lead to smaller cerebellum. To test this, we

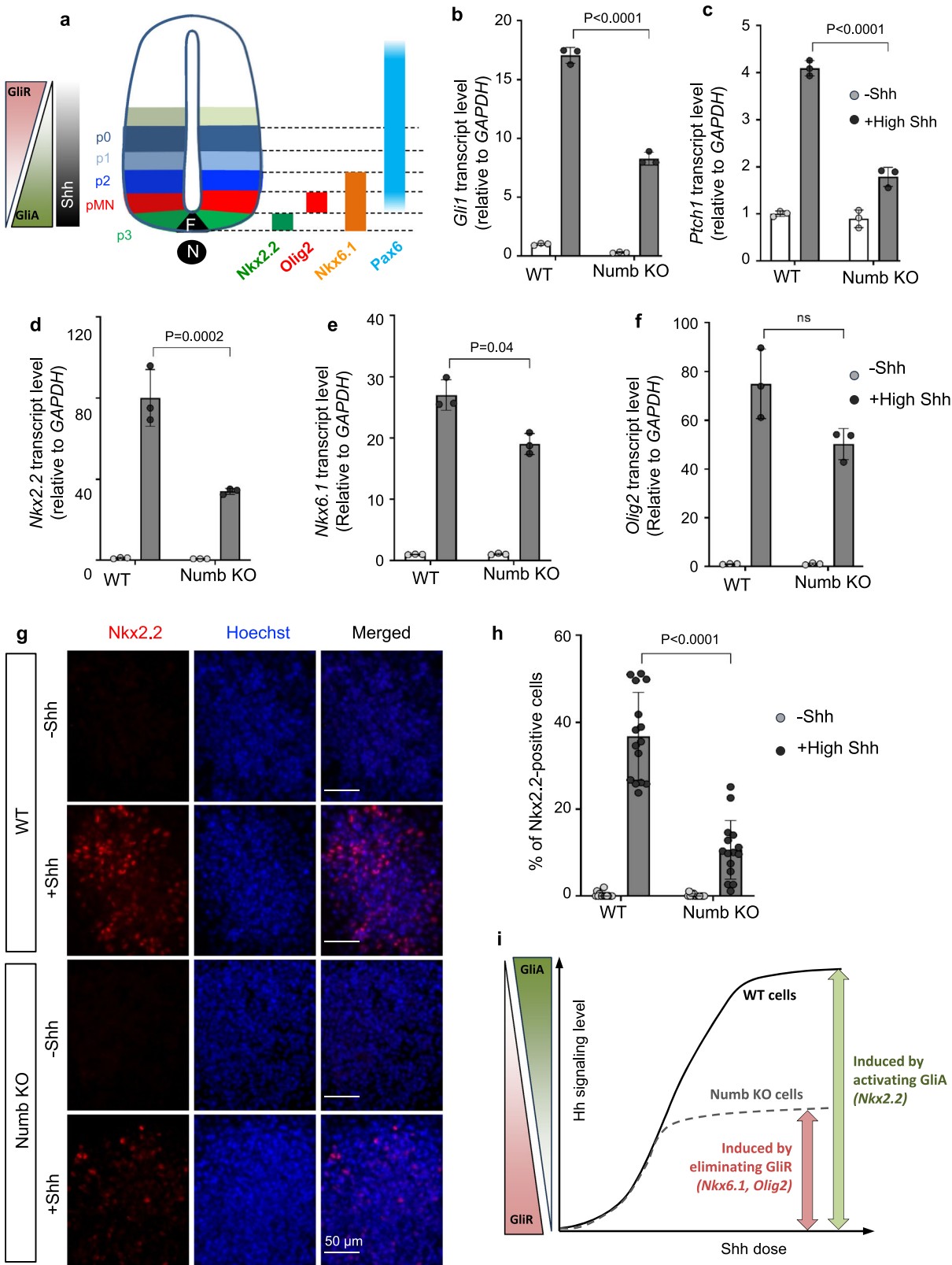

measured the area of the EGL, which is proportional to the size of the GCP population, and the overall cerebellar area in the developing cerebellum. We analyzed sagittal sections from P6 cerebellum at the same mediolateral levels. We found that both the EGL area and the overall cerebellar area are significantly reduced in the cDKO mice compared to control mice (Fig. 8j, k). Importantly, *Numbl* single knockout had no significant impact on the overall cerebellar size at P6

(Supplementary Fig. 15b). Together with our results in NIH3T3 cells, these results suggest that Numb likely plays a more important role than Numbl in GCP proliferation.

After their proliferative phase, GCPs differentiate into mature granule neurons, exit the EGL and migrate inwards to populate the internal granule layer (IGL). To determine whether the reduced GCP proliferation impacts the overall outcome of granule neurons, we

**Fig. 7 | Numb is required for high-level Hh responses in mouse spinal cord neural progenitor cells (NPCs). a** Diagram illustrating embryonic spinal cord progenitors and their associated markers (adapted from Ribes et al. [49]). The bars on the right side depict the specific transcription factors in their respective domains. N notochord, FP floor plate, pMN motor neuron progenitors; p0, p1, p2, and p3, ventral interneuron progenitors. **b, c** Hh signaling activity was assessed by the transcript levels of Hh-target genes, *Gli1* and *Ptch1*, in WT and *Numb* KO mouse spinal cord NPCs. NPCs underwent differentiation in two conditions: either retinoic acid (RA) without Shh, or RA with 100 nM Shh (+High Shh) for 3 days. **d–f** Transcription levels of specific transcription factors induced by high Shh in WT and *Numb* KO NPCs. Cells were differentiated in RA-containing medium without or with 100 nM Shh for 3 days. Three independent differentiation experiments were performed with similar results. *n* = 3 biological replicates. Statistics in (**b–f**): Two-way ANOVA with multiple comparisons (Tukey test). Data are shown as mean ± SD.

**g** Representative images of NPC differentiation into Nkx2.2-positive NPCs after 3 days of induction. **h** The percentage of cells positive for Nkx2.2 in WT or *Numb* KO NPCs. *n* = 15 images from three independent experiments were quantified for each condition. Each image represents one NPC colony consisting of approximately 200–300 cells. Data are shown as mean ± SD. Statistics: Two-way ANOVA with multiple comparisons (Tukey test). **i** Schematics of Hh signaling and NPC differentiation. NPC cell fate in the spinal cord is ultimately determined by the opposing gradient of Gli activator (GliA) and Gli repressor (GliR). In *Numb*-null ES cells, even with high Hh stimulation, GliA activation is impeded, but cells are still able to curb Gli3R production. Consequently, *Numb*-null ES cells fail to differentiate into cell fates that depend on GliA (Nkx2.2-positive NPC) but are able to differentiate into cell fates that are reliant on Gli3R elimination (Olig2- or Nkx6.1-positive NPCs). Source data are provided as a Source Data file.

measured the area of the IGL in cerebellar sections at the same mediolateral levels in adult mice. Our results revealed a significant reduction in the IGL area in cDKO mice compared to control mice (Supplementary Fig. 15c–e). Furthermore, the total area of the cerebellum in adult mice was also significantly reduced in cDKO mice compared to control mice. Although the IGL area was diminished, the organization of granule neurons did not appear to be affected. These results demonstrate that the reduction in GCP proliferation and cerebellar size observed at P6 is not simply a delay in development. Finally, the total body weight of adult cDKO mice is comparable to that of control mice (Supplementary Fig. 15f), indicating that the reduced cerebellar size is not due to an overall decrease in the body size of cDKO mice.

In summary, Numb loss diminishes Hh signaling in GCPs, resulting in decreased GCP proliferation and reduced cerebellar size, while the overall organization of the cerebellum remains largely unaffected. Thus, our in vivo results in the cerebellum corroborate our findings in NIH3T3 cells and NPCs that Numb is required for the maximal activity of Hh signaling.

## Discussion

In this study, our TurboID-mediated ciliary proteomics revealed a distinct group of cilium proteins with only partial overlap with results from previous cilium proteomics[19–21]. To increase the chance of capturing discrete cohorts of proteins, we first leveraged TurboID, a proximity labeling enzyme with distinct mechanisms from APEX. TurboID is derived from the biotin ligase BirA[61], and mediates biotin tagging to primary amines, such as the lysine sidechain. In contrast, APEX is a derivative of ascorbate peroxidase that catalyzes the tagging of biotin-phenol to electron-dense residues such as tyrosine[62]. These distinct labeling mechanisms allow them to target different cohorts of proteins. Second, we targeted TurboID to the juxtamembrane domain by fusing it to truncated Arl13b (90aa). These factors maximized our chance of capturing signal transducers that are recruited to the juxtamembrane region during signaling transduction.

The ciliary pocket is associated with the endocytosis of ciliary proteins[36,37,63]. However, there has been a lack of studies elucidating how this is achieved at the molecular level. In this study, we identified Numb as part of an endocytic machinery at the ciliary pocket that plays an important role in the control of Ptch1 protein levels in the cilium. We showed Numb's localization at the ciliary pocket (Fig. 3a–c), and determined that it interacts with Ptch1 via its PTB domain and N-terminus. The C-terminus of Numb interacts with components of the endocytic and vesicle sorting machinery, such as α-adaptin and EHD family members[42,43,64–67]. Through these molecular interactions, Numb incorporates Ptch1 into CCVs, thereby facilitating Ptch1 departure from the cilium (Fig. 3d–i). This Numb-mediated endocytosis of Ptch1 may also occur at the resting state, but this process is significantly enhanced upon the binding of Shh to Ptch1 (Fig. 3h, i). It is possible that upon binding to Shh, Ptch1 adopts a conformation that promotes its

interaction with Numb. This enhanced interaction could facilitate the efficient incorporation of Ptch1 into CCVs, thereby promoting its internalization from the ciliary pocket. The clearance of Ptch1 from the cilium is essential for the maximal activation of the Hh pathway (Figs. 5–6). In cells lacking a ciliary pocket, we speculate that Ptch1 endocytosis may occur at the cytoplasmic membrane surrounding the base of the cilium.

Among all the Numb truncated variants, NPTB exhibits the highest percentage of localization at the cilium (86.4%) (Supplementary Fig. 5a). This is likely due to the absence of C-terminal motifs known to bind to adaptins[42,43,64–66]. As a result, NPTB cannot be incorporated into CCVs, leading to its retention in the ciliary pocket. This implies that full-length Numb localizes to the ciliary pocket in a highly transient and dynamic manner. Numb is integrated into the endocytosis machinery and exits the ciliary pocket immediately after it reaches the pocket. Therefore, the level of endogenous Numb at the ciliary pocket is low, and we speculate that it is below the detection threshold of immunostaining with anti-Numb antibodies. A similar phenomenon has been noted with other ciliary proteins, such as PKA. Multiple PKA subunits were detected in the cilium in mass spectrometry studies[21]. Further, when PKA subunits were fused to EGFP, their presence was observed in the cilium[21,68]. However, efforts to detect endogenous PKA in the cilium in WT cells using immunostaining methods have not yielded successful results.

We found that Numb positively regulates Hh signaling by facilitating Ptch1 exiting from the cilium upon Shh stimulation (Fig. 5). Our results differ from a previous report showing that Numb negatively regulates Hh signaling by facilitating the degradation of Gli1[44]. Multiple lines of evidence support the idea that our results reveal the physiological role of Numb in Hh signaling.

First, most experiments in the previous study[44] were conducted in non-ciliated cell lines, and hence may not be relevant to the physiological process of Hh signal transduction in the vertebrate system. Second, we found that diminished Hh signaling is rescued in *Numb* KO cells by: (1) restoring Numb expression, (2) stimulating cells with a Smo agonist; (3) expressing a constitutively active Smo mutant, (4) suppressing ciliary PKA with PKI, and (5) knockdown of Sufu. These results show that Numb regulates Hh signaling upstream of Smo, consistent with our identified role of Numb in Ptch1 internalization from the ciliary pocket. Third, in the developing cerebellum where Hh signaling plays a mitogenic role, genetic depletion of *Numb* leads to reduced GCP proliferation. This result provides strong evidence that Numb acts as a positive regulator in Hh signaling under physiological conditions. Finally, we found that ablation of Numb markedly reduced Gli1 protein levels, most likely as a consequence of the reduced *Gli1* transcription. *Gli1*, as a target gene of the Hh pathway and an amplifier of Hh signaling, is induced exclusively after Hh signaling is activated. Gli1 protein levels in a resting state are undetectable. Hence, the scenario described in the previous study may potentially be relevant to specific circumstances, such as certain cancer cells where Hh signaling is

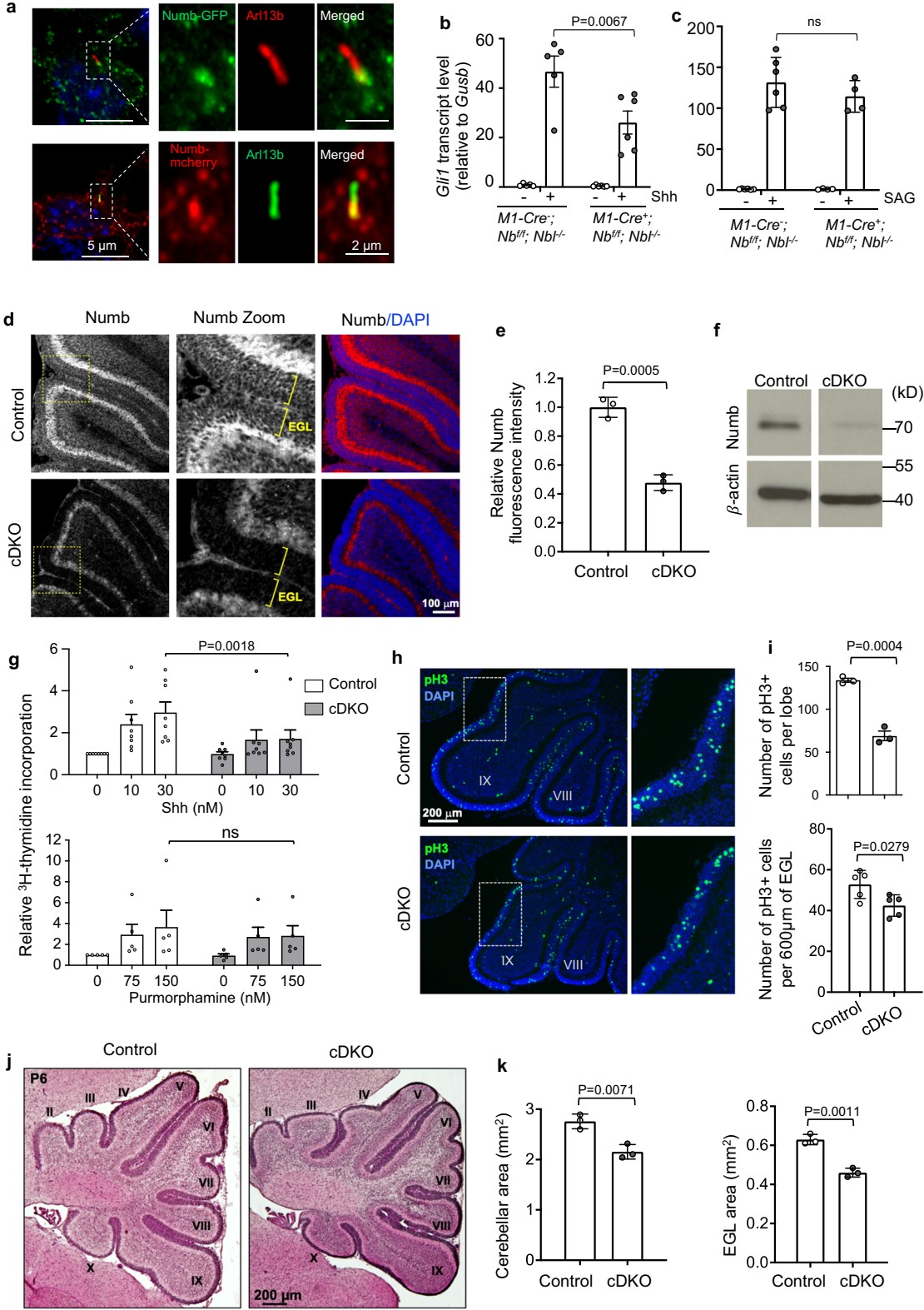

triggered by mutations downstream of Ptch1. However, it may not be applicable to the physiological process of Hh activation.

Our results show that in cells lacking Numb, Ptch1 remains in the cilia in the presence of Shh. This permits a moderate level of Hh signaling activation but blocks the attainment of maximal Hh signaling levels. These results are consistent with several previous studies, highlighting that when Ptch1 clearance from the cilium is hindered,

maximal activation of Hh signaling is compromised[16,69,70]. One explanation is that a Ptch1-free ciliary environment is needed for the maximal activation of Hh signaling. Ptch1 suppresses Hh signaling by preventing the access of Smo to free cholesterols in the cilium, a step required for Smo activation. Shh binding blocks Ptch1 activity as a cholesterol transporter[8–12]; however, when Ptch1-Shh complex persists in the cilium, the ciliary environment is not conducive to optimal

**Fig. 8 | Numb is required for Hh signaling and Shh-induced proliferation in GCPs. a** Numb localizes to the bottom section of the primary cilium in cultured GCPs. GCPs infected with lentivirus expressing Numb-GFP or Numb-mCherry were co-stained with Arl13b. Experiments were performed three times with similar results. **b, c** Hh signaling activity in GCPs isolated from P6 mice. GCPs were cultured in the presence of 1 nM Shh (**b**) or 150 nM SAG (**c**) for 20 h. mRNA was isolated and *Gli1* mRNA levels were measured by qRT-PCR. *n* = 4–6 animals per condition. *M1-Cre, Math1-Cre; Nb, Numb; Nbl, Numblike*. **d** Sagittal sections of cerebellum from P6 control and *Numb;Numbl* cDKO mice immunostained for Numb. The immuno-fluorescence intensity of Numb in the EGL of cDKO cerebellum is lower than the control cerebellum. Boxes indicate the zoomed regions. Brackets denote the EGL. **e** Quantification of the mean Numb fluorescence intensity in the EGL. *n* = 3 cerebella per group. **f** Western blotting showing that Numb protein is reduced in GCPs isolated from cDKO cerebellum. **g** GCPs isolated from P6 cerebellum were cultured for 48 h in the presence of Shh or Purmorphamine. GCP proliferation was measured by ³H-thymidine incorporation. *n* = 8 experiments (Shh) and 5 experiments (Purmorphamine), and each condition included 1–5 animals. **h** Sagittal sections of cerebellum at the same mediolateral levels from P6 control and cDKO mice immunostained for the mitotic marker pH3. Boxes indicate zoomed regions. **i** Quantification of the number of pH3-positive cells in the EGL in one lobe of the cerebellum, and pH3-positive cells per 600 µm of the EGL. *n* = 3 or 5 cerebella per group. **j** Sagittal sections of cerebellum at the same mediolateral levels from P6 control and cDKO mice, stained with H&E. Roman numerals indicate lobules. **k** Quantification of the cerebellar area and EGL area in P6 mice measured at the same mediolateral level. *n* = 3 cerebella per group. All results are shown as mean ± SEM. Statistics in (**e, i, k**): two-tailed Student's *t*-test. Statistics in (**b, c, g**): Two-way ANOVA. ns not significant. Source data are provided as a Source Data file.

activation of Smo. Therefore, in Numb-loss cells, even when Smo accumulates in the cilium, Smo is not fully activated. The ciliary Smo may adopt an inactive conformation similar to that induced by cyclopamine[48]; alternatively, only a subpopulation of Smo in the cilium may be activated. Under either condition, downstream signaling is activated sufficiently to cease Gli3R production but not robustly enough to enable Gli2 activation (Fig. 6). This observation is further substantiated by our fundings in NPCs, where Numb loss allowed cell differentiation into Olig2- and Nkx6.1-positive progenitors that rely on medium-to-low Hh activity, but blocked differentiation into cell fates that require high Hh signaling, such as Nkx2.2-positive progenitors. Notably, our results in NPCs align with the reported phenotype in the spinal cord of *Numb* cKO;*Numblike* KO mice, where Olig2-positive progenitor cells appear to be correctly specified[25].

In the developing cerebellum, Numb regulates granule cell maturation and Purkinje cell maturation[71] and is required for BDNF-mediated migration of GCPs after they exit the cell cycle[59]. We found that the cerebellum of *Numb;Numbl* cDKO mice (Fig. 8) is smaller, as a result of reduced GCP proliferation. Consistent with a role for Numb in granule cell maturation and proliferation, conditional removal of *Numb* with En2-Cre, a Cre line that ablates *Numb* from the entire hindbrain at early developmental stages, also results in a smaller cerebellum[71].

In a previous study[16], Ptch1 levels at the cell surface were regulated by Smurf-mediated ubiquitination and the ensuing degradation in the endosome. Although Ptch1 levels in the cilium increased after Smurf knockdown, this could potentially result from the overall elevated Ptch1 levels in the cytoplasmic membrane. Nevertheless, consistent with our observations in *Numb*-null cells, Ptch1 persists in the cilium even after its binding to Shh in Smurf-deficient cells, leading to attenuated Hh signaling. The Smurf-mediated Ptch1 degradation may operate in conjunction with our current model of Ptch1 exit from the cilium: Numb facilitates the endocytosis of Ptch1 at the ciliary pocket by incorporating Ptch1 into CCVs, and the fate of endocytosed Ptch1 is determined by its ubiquitination status. Collectively, the findings from both Yue et al.[16] and our study underscore the critical significance of clearing Ptch1 from the cilia for the maximal activation of Hh signaling. Our discovery of Numb as a positive regulator of Hh signaling expands our understanding of Hh signaling and reveals an endocytosis machinery at the ciliary pocket that could play crucial roles in the regulation of ciliary protein levels.

## Methods
### Antibodies used in this study
The following antibodies were used in this study: Mouse anti-acetylated tubulin (Sigma, Cat#: T6793, 1:1000 for IF, RRID: AB_477585), Rabbit anti-Arl13b (Proteintech, Cat#: 17711–1-AP, 1:300 for IF, 1:2000 for WB, RRID: AB_2060867), Rat anti-Arl13b (BiCell Scientific, Cat#: 90413), Rabbit anti-Smo (Matthew Scott lab, Stanford University, 1:500 for IF, https://doi.org/10.1126/science.1139740), Rabbit anti-Ptch1

(Matthew Scott lab, Stanford University, 1:500 for IF, 1:2000 for WB, https://doi.org/10.1126/science.1139740), Mouse anti-Ptch1 (Santa Cruz Biotechnology, Cat#: sc-293416), Goat anti-Gli2 (R&D Systems, Cat#: AF3635, 1:200 for IF, RRID: AB_2111902), Goat anti-Gli1 (R&D Systems, Cat#: AF3455, 1:1000 for WB, RRID: AB_2247710), Mouse anti-PDGFR-α (Santa Cruz Biotechnology, 1:1000 for WB, Cat#: sc-398206), Chicken anti-GFP (Aves labs, Cat#: GFP-1020, 1:500 for IF, RRID: AB_10000240), Rabbit anti-GFP (Thermo Fisher Scientific, Cat#: A-11122, RRID: AB_221569), Rabbit anti-Numb (Cell Signaling Technology, Cat#: 2756, 1:200 for IF, 1:2000 for WB, RRID: AB_2154298), Rabbit anti-Numb (Abcam, Cat#: ab14140, RRID: AB_443023), Rabbit IgG (Cell Signaling Technology, Cat#: 2729, RRID: AB_1031062), Rabbit anti-HA (Cell Signaling Technology, Cat#: 3724, 1:200 for IF, 1:2000 for WB, RRID: AB_1549585), Mouse anti-GAPDH (Thermo Fisher Scientific, Cat#: MA5-15738, 1:2000 for WB, RRID: AB_10977387), Mouse anti-NKX2.2 (Developmental Studies Hybridoma Bank, Cat#: 74.5A5, RRID: AB_2314952, AB_531794), Mouse anti-Olig2 (Aves labs, Cat#: OLIG2-0100, RRID: AB_2924438), Mouse anti-V5 (Thermo Fisher Scientific, Cat#: R960-25, RRID: AB_2556564), Mouse anti-β-actin (Sigma, Cat#: A5441, RRID: AB_476744), Rabbit anti-pH3 (Sigma, Cat#: 06-570, RRID: AB_310177), Rat anti-Brdu (Abcam, Cat#: ab6326, RRID: AB_305426), Alexa Fluor 647 Streptavidin (Jackson ImmunoResearch Labs, Cat#: 016-600-084, RRID: AB_2341101), Donkey anti-Mouse Rhodamine (Jackson ImmunoResearch Labs, Cat#: 715-025-151, RRID: AB_2340767), Donkey anti-Rabbit AlexaFluor 488 (Jackson ImmunoResearch Labs, Cat#: 711-545-152, RRID: AB_2313584), Donkey anti-Mouse AlexaFluor 647 (Jackson ImmunoResearch Labs, Cat#: 715-605-151, RRID: AB_2340863), Donkey anti-Chicken AlexaFluor 488 (Jackson ImmunoResearch Labs, Cat#: 703-545-155, RRID: AB_2340375), Donkey anti-Goat AlexaFluor 488 (Jackson ImmunoResearch Labs, Cat#: 705-545-003, RRID: AB_2340428), Donkey anti-Rabbit AlevaFluro 647 (Jackson ImmunoResearch Labs, Cat#: 711-605-152, RRID: AB_2492288), Donkey anti-Goat AlexaFluro 647 (Jackson ImmunoResearch Labs, Cat#: 705-605-147, RRID: AB_2340437), HRP-Conjugated Streptavidin (Thermo Fisher Scientific, Cat#: N100).

### Cell line generation, cultivation, and manipulation
NIH3T3 cells (Cat#: CRL-1658) and 293 T cells (Cat#: CRL-3216) were purchased from ATCC. Cells were cultured in DMEM (supplemented with 10% calf serum). Ciliation was induced by reducing the growth media to 0.5% calf serum for 24 h. Transfections were performed using Lipofectamine 2000 (Invitrogen) according to manufacturer's guidelines. *Numb* gene was disrupted in NIH3T3 cells using CRISPR/Cas9-mediated genome editing with gRNAs targeting exon 1. Cell line clones were obtained by single-cell sorting. Clones with the disrupted gene were screened by immunofluorescence (IF) and Western blotting for the target protein using protein-specific antibodies. Selected positive clones were further characterized by sequencing, confirming missense mutation leading to early termination of translation and frameshift mutation. NIH3T3 cell lines stably expressing Cilium-TurboID and Non-

cilium-TurboID were generated using lentivirus plasmids encoding Cilium-TurboID and Non-cilium-TurboID and confirmed by sequencing. To induce Shh signaling, growth media were supplemented with either 100 nM SAG or ShhN conditioned medium (20–30% [vol/vol] depending on batch) produced with 293 EcR-Shh-N cells. The 293 EcR-Shh-N cell line is a gift from Dr. Rajat Rohatgi lab, Stanford University[52].

## TurboID labeling experiments and subcellular fractionation
For biotin labeling, cells were incubated in the presence of 50 µM biotin for 10 min. For non-labelled samples, water was added instead of biotin. After 10 min of incubation at 37 °C, the medium was aspirated quickly, and cells were washed three times with 1× PBS. For fluorescence microscopy, cells were immediately fixed. For proteomic and Western blot analyses, cells were lysed with subcellular fractionation buffer (20 mM HEPES, 10 mM KCl, 2 mM MgCl2, 1 mM EDTA, 1 mM EGTA, 1 mM DTT, pH 7.5 and protease inhibitors) and by 15 passages through a 27-G needle. Nuclei were separated from the post-nuclear supernatant (PNS) by centrifugation at $720 \times g$ (3000 rpm) for 5 min The PNS was also respun at 8000 rpm (10,000 × g) for 5 min and then brought to Lysis Buffer (0.5% NP-40, 0.1% SDS, 0.5% sodium deoxycholate, 150 mM NaCl, 25 mM Tris/HCl, pH 7.5, and protease inhibitors), sonicated and clarified by centrifugation at 10,000 g for 10 min.

## Streptavidin capture
After determining the protein concentrations of lysates from the TurboID labeling experiments, they were adjusted to equal concentrations and volumes as starting material, from which samples were taken as loading control for SDS-PAGE and Western blot analysis. The remaining lysates were added onto washed and equilibrated streptavidin magnetic beads, and biotinylated proteins were allowed to bind for 1.5 h at room temperature. Beads with bound proteins were washed extensively with a series of buffers to remove nonspecific binders: twice with Lysis Buffer, once with 1 M KCl, once with 0.1 M Na2CO3, once with 2 M Urea in 10 mM Tris-HCl, pH 8.0, twice with Lysis Buffer, and finally with 1× PBS. For Western blot analysis, the beads were eluted with 2 × SDS sample buffer.

## On-beads trypsin digestion of biotinylated proteins and TMT labeling
Beads with bound proteins were incubated in 100 mM TEAB and reduced using 10 mM DTT at 55 °C for five minutes, followed by 25 min at room temperature with head-over-head mixing. The reduced proteins were alkylated using 30 mM acrylamide for an additional 30 min of room temperature with head-over-head mixing. Peptides were digested using trypsin/LysC (Promega) overnight with head-over-head mixing. Peptides were quenched with 1% formic acid and separated from the beads; 0.1% formic acid was used as a second elution prior to C18 microspin column clean up and speedvac to dryness. Peptides were quantified using the Pierce Quantitative Fluorometric Peptide Assay (Thermo) to match input peptide amounts for TMT labeling. Aliquots of each sample were pooled for a common reference prior to isobaric labeling. Peptides were resuspended in 100 mM TEAB and tagged as described in the manufacturer's protocol. Tagged peptides were then pooled to generate a multiplexed sample. This sample was then fractionated offline using high pH reverse-phase fractionation to increase peptide observations.

## Mass spectrometry
In a typical mass spectrometry (MS) experiment, dried, desalted, and labeled peptides were reconstituted in 2% aqueous acetonitrile. MS experiments were performed using liquid chromatography (LC) with an Acquity M-Class UPLC (Waters) followed by MS using an Orbitrap Fusion Tribrid MS (Thermo Scientific). A flow rate of 300 nL/min was used, where mobile phase A was 0.2% (v/v) aqueous formic acid and mobile phase B was 0.2% (v/v) formic acid in acetonitrile. Analytical

columns were prepared in-house, with an internal diameter of 100 µm packed with NanoLCMS solutions 1.9 µm C18 stationary phase to a length of approximately 25 cm. Peptides were directly injected into the analytical column using a gradient (2% to 45% B, followed by a high-B wash) of 120 min for each fraction. MS was operated in a data-dependent fashion using Collision-Induced Dissociation (CID) fragmentation for MS/MS spectra generation collected in the ion trap on the Fusion with MS3 detection of the reporter ions using synchronous precursor selection (SPS) to improve measuring accuracy.

For proteomic data acquisition, ions were generated by nanoelectrospray ionization with a 1600V spray voltage applied to the emitter, approximately 2 mm from the entrance capillary, which was set at 300 °C for efficient dissolution. The instrument method was consistent throughout the 120-minute run, with a 2 s nominal cycle time. MS1 was collected in the orbitrap at a resolution of 120 K, with an AGC target of 400 k, across a scan range of 450–2000 m/z. MS2 was operated under typical data-dependent conditions, with a dynamic exclusion of 45 s following a single observation for peptides with charge states between 2 and 5. MS2 isolation windows were 1.2 m/z, with CID used for fragmentation and detection by ion trap. MS3 was performed using HCD activation and synchronous precursor selection for peptide fragments between 400–1200 m/z for TMT quantification. MS3 isolation used 7 notches, and windows of 2 m/z, with a detection range of 100–500 m/z in the orbitrap at 60 K resolution.

Prior to SPS MS3, high pH reversed-phase offline fractionation was performed following the protocol of the Pierce Fractionation Kit (Cat # 84868). Briefly, samples were loaded onto an equilibrated, high-pH, reversed-phase fractionation spin column. Peptides were bound to the hydrophobic resin under aqueous conditions, desalted by water washing and low-speed centrifugation. A step gradient of increasing acetonitrile concentrations in a volatile high-pH elution solution was then applied to the columns to elute bound peptides into eight different fractions collected by centrifugation. Each fraction was then dried in a vacuum centrifuge and stored until analysis by mass spectrometry.

## MS data processing
For data analysis, the .RAW data files were checked using Preview (Protein Metrics) to verify calibration and quality prior to further analysis. Data were then processed using Proteome Discoverer v1.5 (Thermo) for quantitative analysis of reporter ion ratios, and using the Byonic (Protein Metrics) node to identify peptides and infer proteins against the *Mus musculus* database from Uniprot (including isoforms and concatenated with common contaminant proteins). Proteolysis with Trypsin/LysC was assumed to be specific with up to two missed cleavage sites, and allowing for common modifications. Precursor mass accuracies were held within 12 ppm with fragment ions held within 0.4 Da (for CID) and 12 ppm (for HCD). Proteins were held to a false discovery rate of 1%, using standard approaches[72].

## Numb KO mESC generation and neural progenitor differentiation
The mouse embryonic stem cells were a gift from the Briscoe Lab. The original HM1 mouse embryonic stem cells were purchased from Thermo Fisher Scientific (A34959). The *Numb* gene was disrupted in a stable Gli-binding site (GBS)-Venus Hedgehog signaling reporter line (DV12) generated in HM1 mouse embryonic stem cells (mESCs)[52]. To ablate Numb function, CRISPR/Cas9-mediated genome editing was used. Briefly, two sequences targeting exon 1 of *Numb* (5'-GAAA-GACGTTTATGTCCCAG-3' and 5'-GGAAGCTACACTTTCCAGTG-3') were individually cloned into plasmid pSpCas9(BB)−2A-Puro (PX459) V2.0 (Addgene #62988). These guide constructs were then electroporated into mESCs using the Lonza Nucleofector 2b Device and Cell Nucleofector Kit (Lonza #VAPH-1001). mESCs were cultured in feeder-free 2i media [DMEM/F-12 (Gibco #11320033) and Neurobasal Medium

(Gibco #21103049) (prepared at a 1:1 ratio) supplemented with N-2 Supplement (Gibco #17502048), serum-free B-27 Supplement (Gibco #17504044), penicillin/streptomycin (Gibco #15140122), glutaMAX (Gibco #35050061), bovine serum albumin (Thermo Scientific Chemicals #AAJ64248AE), 55 µM 2-mercaptoethanol (Gibco #21985023), 3 µM CHIR99021 (Axon #1386), 1 µM PD 98059 (Tocris Bioscience #1213), and ESGRO recombinant mouse LIF protein (1000 units/ml, MilliporeSigma #ESG1107)]. Antibiotic selection was initiated 24 h after nucleofection in 2i media containing 1.5 µg/ml puromycin (Fisher BioReagents #BP2956100) for 72 h. Clones were picked individually, expanded, and genomic DNA was collected using QuickExtract DNA extraction solution (Lucigen, #QE09050). The region surrounding the targeted site was PCR-amplified (F: 5'-agggtttggggtgggtttt-3' and R: 5'-gctttgtctgggttccttcc-3') to visualize the successful deletion facilitated by the guides. A loss of full-length Numb protein was then verified by Western blotting using protein-specific antibodies.

The mESCs were differentiated into spinal neural progenitors using the protocol in Pusapati et al. [52], which was modified with minor modifications from Gouti et al. [51]. To prepare the mESCs for differentiation, the mESCs were passaged twice on CF1 mitomycin C-treated mouse embryonic fibroblasts (MitC-MEFs, Gibco Cat #A34959) in mESC media [DMEM containing high glucose (Cytiva #SH30081.FS), 15% fetal bovine serum (Gibco #A5256701), MEM non-essential amino acids (Gibco #11140076), penicillin/streptomycin (Gibco #15140122), glutaMAX (Gibco #35050061), EmbryoMax nucleosides (MilliporeSigma #ES008D), 55 µM 2-mercaptoethanol (Gibco #21985023), and ESGRO recombinant mouse LIF protein (1000 units/ml, MilliporeSigma #ESG1107)]. For spinal neural progenitor differentiation, the feeders were first removed from the mESCs. Removal of MitC-MEFs was accomplished by first lifting all the cells off of the plate using 0.25% trypsin/EDTA (Gibson #25200072), inactivating the trypsin using mESC media (supplemented with no LIF), and then incubating the cells on 10 cm tissue culture plates for two short (20 min) successive periods. To induce differentiation, mESCs were plated onto either coverslips or CellBIND plates (Corning #3335) treated with 0.1% gelatin (Sigma Aldrich Fine Chemicals Biosciences #G139320ML) and cultured in N2B27 media [DMEM/F-12 (Gibco #11320033) and Neurobasal Medium (Gibco #21103049) (prepared at a 1:1 ratio) supplemented with N-2 Supplement (Gibco #17502048), serum free B-27 Supplement (Gibco #17504044), penicillin/streptomycin (Gibco #15140122), gluta-MAX (Gibco #35050061), bovine serum albumin (Thermo Scientific Chemicals #AAJ64248AE), 55 µM 2-mercaptoethanol (Gibco #21985023), and varying components. On Day 0 (the day the cells were plated) and Day 1, the cells were cultured in N2B27 with 10 ng/ml mouse basic fibroblast growth factor (bFGF) recombinant proteins (R&D systems #313FB025/CF). On Day 2, the media was changed to N2B27 with 10 ng/ml bFGF and 5 µM CHIR99021 (Axon #1386). On Day 3, the cells were cultured in N2B27 with either no SHH [100 nM retinoic acid only (RA, Sigma Aldrich Fine Chemicals Biosciences #R262550MG)] or high SHH [100 nM RA and 100 nM SHH). No Shh and High Shh treatments were continued for 3 days before harvesting.

## Immuno-electron microscopy

For correlative light and electron microscopy (CLEM), NIH3T3 cells expressing Numb-HA were plated on gridded-glass bottom dishes (MatTek, cat. no. P35G-1.5-14-C-GRD). After 4% PFA fixation, cells were permeabilized with 0.2% Triton X-100 in PBS for 10 min and underwent immunostaining with rabbit anti-HA (cell signaling, Cat # 3724), follow by the secondary antibody of Alexa 488 FluoroNanogold-conjugated anti-rabbit IgG (Nanoprobes, #7204). The nanogold signal was enhanced via the GoldEnhance EM Plus kit (Nanoprobes, #2114). After that, fluorescence and DIC images were taken with a Leica DMi8 microscope. The position of cells was recorded using grid numbers on cover glasses. We used a modified protocol from UC Berkeley EM facility to process the cells. Briefly, cells were post-fixed in 1% OsO$_4$

with 1.6% potassium ferricyanide (K$_3$Fe(CN)$_6$) in PBS for 30 min. Cells were then rinsed with 1x PBS 3 times for 5 min each and briefly rinsed with distilled H$_2$O. To dehydrate the cells, an increasing percentage of 200-proof ethanol was added to the plates in 10-minute increments: 30%, 50%, 70%, 95%, 100%, 100%, 100%. Next, cells were infiltrated with increasing resin amounts to ethanol for 30 min each; 1:4, 1:2, 1:1, 2:1. Finally, cells were incubated with pure resin 3 times for 30 min each. At the end of incubation, the remaining resin was removed, and a thin layer of pure resin was added to cover the well-containing cells. The samples were then incubated at 60 °C for 16 hrs and microtomed for imaging.

## Immunofluorescence and microscopy

Cells were grown on round 12 mm #1.5 coverslips and fixed in 4% PFA for 10–15 min at room temperature. After fixation, cells were permeabilized and blocked in blocking buffer (2% donkey serum, 0.2% triton X-100, in PBS) for 1 h at room temperature. After blocking, cells were incubated with primary antibody diluted in blocking buffer for 1 h at room temperature or 4 °C overnight, washed three times with 1× PBS over 15 min, and incubated with Alexa Fluor 488-, rhodamine-, or Alexa Fluor 647-coupled secondary antibodies (Jackson ImmunoResearch) in blocking buffer for 1 h, and then incubated in Hoechst (1 µg/ml in PBS) for 10 min. Finally, cells were washed five times in PBS and mounted on glass slides using Fluoromount G (Southern Biotech; 0100-01). Prepared specimens were imaged on a Leica DMi8 (LAS X software) with Plan Apochromat oil objectives (63x, 1.4 NA) or a LSM880 confocal microscope (Zeiss). Images were processed using ImageJ.

## Expansion microscopy

The gelation and proteinase digestion steps were similar to the proExM protocol by Tillberg et al. [40] with two modifications: (1) after cells were fixed and stained with primary antibodies, we added 4%PFA to cross link the antibody to the samples; and (2) we stained the sample with secondary antibody after proteinase digestion, with the intention to protect the fluorescent dyes.

Briefly, fixed cells were incubated in primary antibodies for 2 h at room temperature or at 4 °C, overnight and then were fixed in 4% PFA for 20 min. After fixation, cells were incubated in the 0.1 mg/ml AcX (Acryloyl-X SE, Invitrogen, cat. no. A20770) solution for 2 to 3 h at room temperature. Then cells were polymerized in the gelling solution by mixing Stock X (8.6% Sodium acrylate, 2.5% Acrylamide, 0.15% N,N'-Methylenebisacrylamide, 11.7% Sodium chloride, 1x PBS), water, 10% TEMED stock solution, and 10% APS stock solution in a 47:1:1:1 ratio. The gel was then digested with proteinase K (NEB, cat. no. P8107S) at a final concentration of 8 U/ml in digestion buffer (50 mM Tris, pH 8.0, 1 mM EDTA, 0.5% Triton X-100, and 0.8 M NaCl) for 12 h at room temperature. After digestion, proteinase K was removed by four washes with excessive PBS (10 min each time) and the gel was incubated in the secondary antibody for 2 h at room temperature. The post-expansion labeled hydrogel was then washed and expanded by four washes with excessive water (at least 30 min each time) and mounted with superglue. Imaging was performed by a AiryScan LSM880 confocal microscope (Zeiss) with a 63x lens.

During imaging, the elastic nature of the hydrogel makes it difficult to complete a z-stack optical scanning. The short working distance of the lens presses the hydrogel, dislocates the samples' position and causes image blurring. To avoid the potential artifacts of imaging blur, we captured one single optical plane focusing only on the ciliary pocket.

## Quantification of cilium length and protein intensity in the cilium

Cilium length was measured in Leica LAS X software. A line was drawn along the fluorescent signal corresponding to the ciliary marker, and the length of this line was defined as the length of the primary cilium.

The ciliary protein intensity was measured in ImageJ. Briefly, we first outlined the contour of an individual cilium in the channel of cilium staining. After that, the fluorescence intensity within this contour was read out for each individual channel of the corresponding protein (L1). Finally, the individual cilium contour was manually dragged to the region right next to the cilium, and the intensity within the contour was read out as the background (L2). The final fluorescence intensity for that channel was defined as L = L1−L2.

### Quantification of Ptch1 incorporation in Numb-containing CCVs

The quantification was done in Fiji. All images quantified were taken with a confocal microscope. The region of interest (ROI) was defined as a circle with the cilium base as the center and the cilium length as the radius. Within the ROI, we remove the background by subtracting the global threshold value. Thresholded Manders' Coefficients of Ptch1-YFP were calculated using BIOP JACoP plug-in in ImageJ, based on the co-localization methods described in Manders et al.[73].

### SDS-PAGE and Western blotting

Standard techniques were used for SDS-PAGE and Western blotting. Cells were first washed with PBS, and then were scraped off from the culture surface in RIPA buffer (1% NP-40, 0.1% SDS, 0.5% sodium deoxycholate, 150 mM NaCl, 25 mM Tris/HCl, pH 7.5, and protease inhibitors). Lysates were cleared by centrifugation ($10,000 \times g$ at 4 °C for 15 min), and 25 μg protein was separated on 10% SDS-PAGE gels and transferred onto PVDF membranes. After blocking in 5% BSA, membranes were washed and incubated with primary antibodies and secondary antibodies. Finally, proteins were detected with chemiluminescence substrates (Thermo Fisher Scientific; 34076). Quantitation of bands was performed using ImageJ.

### Animals

All animal work was performed in accordance with the Canadian Council on Animal Care Guidelines and approved by the IRCM Animal Care Committee (animal protocol 2020-02 FC), and the guidelines by the IACUC of the University of California Merced (animal protocol 2023-1156). Mice were maintained in each institute's specific pathogen-free animal facility. The mouse facilities are operated in a 12 light/12 dark cycle, under temperature of 66–72 °F (-20–22 °C) with 40–60% humidity. All mouse lines have been previously characterized: *Math1-Cre* was purchased from the Jackson Laboratory (Strain #: 011104); *Numb/Numbl* double floxed mouse line[74] was obtained from the Jackson Laboratory (Strain #: 005384); *Numbl* null mouse line was from the Cayouette lab[75]. Mice of both sexes (not determined) were randomly used for experiments.

For experiments with conditional deletion of *Numb* and *Numbl*, control mice were *Math1-Cre⁻;Numb^{f/f};Numbl^{f/f}*, *Math1-Cre⁻; Numb^{f/f};Numbl^{f/−}*, or *Math1-Cre⁻;Numb^{f/−};Numbl^{f/−}*. *Numb;Numbl* cDKO mice were *Math1-Cre⁺;Numb^{f/f};Numbl^{f/f}*, *Math1-Cre⁺;Numb^{f/f};Numbl^{f/−}* or *Math1-Cre⁺;Numb^{f/−};Numbl^{f/−}*.

### Histology

Brains were dissected and fixed by immersion overnight in 4% paraformaldehyde (PFA) diluted in PBS. The tissues were then cryoprotected in 30% sucrose, embedded in a sucrose:OCT (1:1) mix and frozen. Histology and immunochemistry were performed based on standard protocols. For immunochemistry on cryosections, brain slices were incubated in blocking solution (2% goat serum, 0.2% Triton X-100 in PBS) for 1 h at room temperature. Subsequently, brain sections were incubated in primary antibodies diluted in blocking solution for 2 h at room temperature or 24 h at 4 °C. Rabbit anti-Numb antibody (Abcam 14140) was used at 1:250. Sections were then washed three times with PBS and incubated with a secondary antibody for 45 min at room temperature. followed by staining with Hoechst 33258 (Sigma). Sections were then counter stained with

Hoechst 33258 and mounted in Fluoromount-G (Fisher) for fluorescence microscopy.

### Isolation of GCPs and in vitro proliferation assays

GCPs were isolated from P6 mice cerebella. Briefly, isolated cerebella were cut into small pieces and treated with trypsin and DNase I. After trituration, single-cell suspensions were centrifuged through a 30% to 65% percoll step gradient. Cells were harvested at the 30% interphase and then resuspended in Neurobasal supplemented with B27, 0.5 mM L-glutamine and penicillin/streptomycin and plated on plates precoated with 100 μg/ml poly-D-Lysine.

For the ³H-thymidine incorporation assay, GCPs were plated at $3 \times 10^5$ cells per well in 96 well plates in triplicate and treated with ShhN for 48 h. GCPs were pulsed with 1 μCi/ml ³H-thymidine 12 h after seeding the cells. The amount of ³H-thymidine incorporation was measured using the Filtermate harvester (PerkinElmer).

### Statistical analyses

Statistical analyses were performed with GraphPad Prism 8. For volcano plot graphs, Student's *t*-test were used, and statistical analyses were performed in Prism 8. Hierarchical cluster analyses were performed according to Ward's minimum variance method.

### Reporting summary

Further information on research design is available in the Nature Portfolio Reporting Summary linked to this article.

### Data availability

The original mass spectra have been uploaded to MassIVE using the identifier MSV000090077 [https://massive.ucsd.edu/ProteoSAFe/dataset.jsp?task=91739d24e0004e27992be536db3d3100]. The dataset can be downloaded from the MassIVE FTP server [ftp://massive.ucsd.edu/v04/MSV000090077/]. This dataset can also be accessed in ProteomeXchange via Identifier PXD035789. All remaining data supporting the findings of this study are available within the paper and its Supplementary Information. Source data are provided with this paper.

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

## Acknowledgements

We thank Dr. Rajat Rohatgi (Stanford University) for the gifts of CRISPR/Cas9 reagents, and Dr. Matthew Scott lab for the antibodies against Ptch1 and Smo. We thank Dr. David Gravano (Stem Cell Instrumentation Foundry, University of California, Merced) for fluorescence-activated cell sorting, and Dr. David Ardell (University of California, Merced) for consultation of proteomic data analysis. We thank Dr. Christopher J. Westlake (Center for Cancer Research, National Cancer Institute) for the gift of EHD1-GFP plasmid. We thank Francois Depault, Christine Jolicoeur and Jessica Barthe for expert technical assistance. We thank Julien Ferent for critical discussions and Kevin Zhang for help with the cerebellum histological analysis. The data in this work was collected, in part, with a confocal microscope acquired through the National Science Foundation MRI Award Number DMR-1625733. Mass spectrometry was performed in the Vincent Coates Foundation Mass Spectrometry Laboratory, Stanford University Mass Spectrometry (RRID:SCR_017801) by Dr. Ryan Leib, Dr. Fang Liu, Kratika Singhal, and Rowan Matney. Correlative ImmunoEM was performed at the Electron Microscope Lab at University of California, Berkeley, under the supervision of Danielle Jorgens. Research in the laboratory of X.G. was supported by NIH/NCI (CA235749), NIH/NIGMS(GM143276), NIH/NCI (CA274595), and NSF CAREER award (IOS-2143711). Research in the laboratory of F.C. was supported by the Canadian Institutes of Health Research (FDN334023 and PJT180637), Brain Canada-Weston Foundation (to F.C. and M.C.), and the Canada Foundation for Innovation (33768 and 39794). F.C. holds the Canada Research Chair in Developmental Neurobiology. Research in the laboratory of J.H.K. was supported by NIH/NIGMS (GM132518) and NIH/NCI (CA015704).

## Author contributions

X. Ge, X. Liu, P. T. Yam and F. Charron conceived the project. X. Ge and X. Liu designed the experiments on cilium proteomics and Numb mechanistic study in cultured cells. X. Liu performed the experiments, analyzed, and interpreted the data on cilium proteomics and the Numb characterization in cultured cells. E. Cai curated and characterized the Numb CRISPR/Cas9 knockout cell lines. O. T. Gutierrez and J. Zhang performed experiments related to quantifications in the cilium length and cilium protein levels in NIH3T3 cells and in primary cultured GCPs. J. H. Kong designed and supervised experiments in NPCs. R. Gen did the CRISPR/Cas9 Knockout in mESC cells and genotyped the colonies. T. Marks performed the NPC differentiation, performed qPCRs and the immunostaining staining and imaging in differentiated NPCs. P. T. Yam, W. J. Chen, S. Schlienger, V. Jimenez Amilburu, V. Ramamurthy, M. Cayouette and F. Charron designed and performed the experiments on *Numb;Numbl* knockout mice, and analyzed and interpreted the data on cerebellar development and GCP proliferation. T. Branon and A. Ting provided the reagents for TurboID and provided technical support for TurboID related proteomics. X. Ge, X. Liu, P. T. Yam and F. Charron wrote the paper.

## Competing interests

Tess C. Branon is an employee of Interline Therapeutics, South San Francisco, CA, USA. The remaining authors declare no competing interests.
