## [Peer Review File · Nature Communications]

Numb positively regulates Hedgehog signaling at the ciliary pocketReviewers' comments:

Reviewer #1 (Remarks to the Author):

This study by Liu et. al. presents a set of experiments in support of the model that Numb acts as a positive regulator of HH signaling by serving as an endocytic adapter for the clearance of PTCH1 from cilia. Numb has already been shown to be involved in Patched endocytosis in the context of axon guidance. The authors seek to establish a role for Numb in “canonical” Hedgehog responses (ie. responses mediated through the GLI transcription factors). The approach used by the authors (proximity biotinylation) has been applied previously to study cilia (and the Hh response within cilia). My assessment is that the novelty and biological impact does not meet the bar for Nature Communications. Most importantly, there is very little *in vivo* evidence that numb serves as a general positive regulator of Hh signaling in embryonic tissues that are classically patterned by Hh ligands (e.g. the neural tube, limb, bone etc). The authors present data from the cerebellum, a tissue where HH ligands drive granule cell proliferation, but in my view they have not conclusively shown that the effects are due to HH signaling (as opposed to other effects of Numb/Numbl). Importantly Numb and Numbl mice have been analyzed extensively in the past (see for example PMID: 12410312). The CNS in these mice have no defects in temporal and spatial patterning (as would be expected of HH defects), but rather in progenitor cell maintenance. To show that Numb/Numbl are involved in Hh signaling *in vivo*, the authors should test the effects of their loss in other Hh-patterned tissues such as the neural tube or limb. This analysis should include both direct measurements of target gene expression and well and assessment of HH-driven patterning and cell differentiation effects.

Major comments:

>>Further characterization of the TurboID cell lines is needed. In Fig.S2: Cilia length, ciliary SMO and ciliary Gli2 should be measured in response to the native ligand SHH (not just the synthetic agonist SAG).

>>The summary graphic in Fig.3C is misleading— there is no evidence that numb specifically co-localizes with clathrin in vesicles derived from the ciliary pocket (as opposed to vesicles derived from the plasma membrane). The authors’ experiments cannot distinguish between these possibilities.

>>Ciliary pocket localization of Numb is based on one marker. It needs to be confirmed by EM.

>>In Figure 3f the co-localization results do not show that PTCH1 is internalized into Numb-containing vesicles. There is some co-localization, but also plenty of PTCH1 puncta that do not co-localize with numb.

>>Does endogenous numb localize in cilia, both in cells and in embryonic tissues where Hh signaling is active (ventral neural tube, limb etc.)? I understand this is technically challenging, but I believe critical to the paper.

>>PTCH1 is cleared from the cilia upon SHH addition. What is the mechanism by which SHH addition

leads to Numb-mediated PTCH1 internalization? Is numb a constitutively acting on PTCH1 or only in response to SHH?

>> The authors call numb an “adaptor” for PTCH1 internalization, yet very few experiments (other than PTCH interaction) show this in a rigorous way. An adaptor links a receptor cargo to the clathrin (or other endocytic) machinery. For example, where does Numb bind to PTCH? Identifying the protein-protein interface that mediates the PTCH1-Numb interaction is critical here. The authors can then make point mutations (in both PTCH1 and Numb) that disrupt the interaction and show that there is an effect on PTCH1 clearance from cilia and Hh signaling. This type of analysis would markedly improve the paper.

>>The effects of Numb loss on HH signaling are partial. In fact, at lower concentration of SHH the effects are minor (e.g. Figure 5b). This should be made clear and discussed, with the explicit statement that these effects are modest.

>>In Figure 4, significant additional work needs to be done to establish that Numb loss impairs PTCH removal from cilia in response to SHH. First, this needs to be done with endogenous PTCH1, not PTCH1 artificially (non-physiologically) elevated after SAG addition. Second, the protein abundance of PTCH1 in cilia should be clearly plotted - and + SHH in both WT and numb ko cells. Plotting a ratio (as the authors have done) is not appropriate since the ratio is generated using a mean value for baseline (and is therefore not a single cilium measurement).

>>In Figure 5, SmoM2 overexpression is used to show that the requirement of Numb is at the level of PTCH1, not SMO. SmoM2 over expression is quite non-physiological, so the authors should test whether Numb inactivation influences signaling by endogenous SMO in response to agonists like SAG and to genetic ablation of negative regulators of signaling downstream of SMO (e.g. GNAS, SUFU).

Reviewer #2 (Remarks to the Author):

In this work, Liu et al. identified Numb as a novel ciliary protein from using an impressive cilia membrane-specific proximity labeling approach. Using cells in culture they found Numb localizes at the ciliary pocket and clathrin-associated vesicles. Experiments in 3T3 cells revealed that similar to the previously reported function of Numb in the non-canonical Shh pathway, Numb mediates the removal of Ptch1 from primary cilia and also that it augments activation of the canonical Shh pathway. They show that knockout or knockdown of Numb in cells partially blocks Ptch1 removal from cilia and compromises Shh signaling. In mice, conditional knockout of Numb along with a null allele of Numb1 in GCPs results in a mild reduction in HH signaling in GCPs and it is claimed to reduce proliferation of GCPs, and it also causes a smaller cerebellum during development and in the adult.

The quality of the data are solid, and most conclusions are appropriate. The findings of this work add to the known roles of Numb in Ptch1 removal by showing a function in the canonical Shh pathway. Overall, the paper will be a valuable Resource paper for cilia and SHH specialists and it also presents a standard

phenotype analysis in the cerebellum of one protein the authors chose to follow up on (Numb). The paper has several limitations for a high impact paper.

Major concerns:

(1) The new findings seem incremental given that two previous papers showed endocytosis is involved in Ptch1 removal from cells. One paper showed that Ptch1 removal from primary cilia is mediated by a Smurf-related endocytosis and subsequent lysosome degradation and this leads to reduced GCP proliferation, and another paper showed that Ptch1 removal from the cell surface in non-canonical SHH signaling involves Numb and endocytosis dependent on Boc.

(2) The Shh/Ptch1-related phenotypes (removal of Ptch1 and GCP expansion) of Numb mutants appear mild, raising the question of whether Numb plays only a minor role in removing Ptch1. Furthermore, the mechanism demonstrated *in vitro* was not validated in a second relevant cell type, such as a human RPE cell line.

(3) It is not clear if the function of Numb on Ptch1 removal from cilia relies on Boc (as in non-canonical signaling) or not, notably Boc is not listed in the ciliary proteome and Izzi et al. showed Boc mutants have a GCP proliferation phenotype. This point needs to be addressed.

(4) The authors should address whether Numb plays a role in cilia formation in GCPs *in vivo* or maintaining the ciliary pocket compartment *in vivo* or *in vitro*.

(5) Does Numb control the protein levels of other SHH signaling-related membrane proteins in GCPs, e.g. is there decreased Boc, Smo accumulation in cilia of mutants after stimulation of SHH signaling?

(6) The phenotype of Math1-Cre; Numb-*fl/fl* mice must be presented to determine whether the loss of Numb contributes to the mild phenotype reported in DCKO.

(7) Numb expression appears to be much higher in Purkinje cells compared to GCPs, but Purkinje cells do not transduce HH signaling. What do the authors think is the function of the high level of Numb in Purkinje cells?

(8) In the mouse experiments, state the genotype of controls. Are they Numb nulls? If not, they should be.

(9) Fig. 7 The number of p3+ cells per lobe cannot be used as a measure of proliferation rate since the DCKO have a smaller cerebellum. The measurement should be p3+Ki67+/Ki67+ cells or p3+ cells per same area of outer EGL, and the same lobule and medial-lateral position used.

(10) "The length of cell cycle was calculated by: 2h x (PCNA+ cells/BrdU+EdU- cells)."

This is not an accurate measure since most of BrdU is cleared by 1 hour. To accurately measure cell cycle length, the authors would need to inject EdU at different times from 12-24 hours after BrdU administration and determine when maximum double labeling occurs.

(11) Fig. 8a, d. Please orient cerebella so dorsal is up and anterior is consistently to left (or right). Midsagittal sections should be shown to be able to interpret the phenotype.

The H&E staining in (d) needs to be improved. Furthermore, in the adult DCKO, a lobule seems to be missing in the central region (L7?). Is this a consistent phenotype?

(12) Bergmann glia respond to SHH and thus must have cilia. Do they express Numb and does it play a role in SHH signaling? A high magnification image should be included with double labeling for Numb and a Bergmann glia marker and/or Purkinje cell marker.

(13) To examine Shh activation in the Numb mutants, besides Gli1 and Ptch1 transcription, the enrichment of Smo and GLI trafficking in cilia should be examined in the mutants, in particular, because if Ptch1 fails to move out of the cilium it has been reported this should block Smo entry into the cilium.

Minor comments:

(1) Fig. 7 The IF staining of Numb in the EGL of the cerebellum seems very low and questionable. Higher magnification images with the negative control (Cre+) are required. Also, qRT-PCR would strengthen the results.

(2) Fig. 6C, D please make it clear what “*” indicates - significant differences between groups or time points?

(3) Fig. 7. Low power images of the cerebella indicating where the high power images are taken from should be included. Also, the lobules the quantification was performed on should be stated.

(4) All data points should be shown in graphs to illustrate variation more accurately than bar graphs.

Reviewer #3 (Remarks to the Author):

In this manuscript, Liu et al use turbo-ID technology to elucidate the protein composition of the cilium in response to Shh treatment and discover that Numb regulates the departure of Ptch1 from the cilium and regulates cerebellar size in mice. This is a highly rigorous and fascinating glimpse into the mechanistic basis of Ptch1 regulation of Shh signaling. I deeply appreciated the NUMB KO/reconstitution data and the SMO2 stimulation data which nicely illustrated where Numb works in the signaling pathway. The proteomic experiments are conducted with the utmost care and the methods used are highly rigorous and state-of-the-art. I only have small suggestions to increase the clarity of the proteomic parts of the paper. The Cre conditional Numb KO data impressively illustrated the in vivo importance of the biochemical findings in the NIH3T3 cells. Overall, it was a fantastic paper.

Minor Criticism:

1) What was the exact distribution of samples into TMT channels and if a second TMT mix was employed, how did it differ from the first? Was the pool channel only used to calculate enrichment or was it also used to expand the TMT experiment into multiple mixes? A supplemental methods figure illustrating the distribution of all samples into TMT channels would help to clarify this point.

2) Typically, high pH reversed phase offline fractionation is employed prior to SPS MS3 due to the long duty cycle for SPS MS3. Was that not done in this case? The methods mention LCMS of each "fraction" but no details of what method was used to fractionate.

3) For the volcano plots (Fig 2d, S3), how were the p values calculated and adjusted for multiple comparison?

4) Fig 2C could be combined with an array of replicate-to-replicate reporter intensity scatter plot in the supplement mentioning the R squared correlation of the replicate pairs on the plots and mentioning the average R squared in the text of the paper. The heatmap does not quantify the correlation quite as rigorously as that although the heatmap does look very impressive.

5) All proteomic data needs to be deposited into the PRIDE database prior to publication and the

proteomic data acquisition parameters should be described in more detail in the methods or supplement. What were the data acquisition parameters on the Fusion?

5) For fig 5d, what was the concentration of Shh used. this information will be helpful in comparison of 5d to 5b

6) possible typo in line 499; impacts seems like it should be impact

Reviewer #4 (Remarks to the Author):

This reviewer was asked to evaluate the ExM imaging. The reviewer finds the data well presented, the method well-described and the immunostaining and results convincing. These are very nice ExM data of cilia. However, the wording is not standard for the description of the data.

Using the term "superresolution microscopy" in the way the authors do in the manuscript in line 214 line 1025, 1034 is in the reviewers opinion not useful. Yes, the expansion results in the resolution of structures that may be separated by distances below the diffraction limit and Airyscan microscopy yields somewhat of a resolution increase, but "superresolution microscopy" is not a technically accurate term. The authors performed Expansion microscopy or they expanded the sample and then they imaged using an airy scan confocal microscope yielding an apparent increase in resolution. They do describe it better later in the manuscript.

Reviewer #1:

This study by Liu et. al. presents a set of experiments in support of the model that Numb acts as a positive regulator of HH signaling by serving as an endocytic adapter for the clearance of PTCH1 from cilia. Numb has already been shown to be involved in Patched endocytosis in the context of axon guidance. The authors seek to establish a role for Numb in “canonical” Hedgehog responses (ie. responses mediated through the GLI transcription factors). The approach used by the authors (proximity biotinylation) has been applied previously to study cilia (and the Hh response within cilia). My assessment is that the novelty and biological impact does not meet the bar for Nature Communications.

We thank the reviewer for these comments. Below we would like to clarify the novelty and significance of our study in the context of previous publications.

- 1) Our study, for the first time, identified an endocytosis mechanism at the ciliary pocket. While it has been speculated that the ciliary pocket is involved in the endocytosis of ciliary proteins, there has been a lack of studies demonstrating how this is achieved at the molecular level. To date, there is no study showing a specific molecule endocytosed from the ciliary pocket. The only direct evidence so far is that at the EM level, clathrin-coated vesicles are enriched in the ciliary pocket^{1,2}. Our study filled this gap by elucidating how the “cargo” molecule Ptch1 is incorporated into the endocytic machinery at the ciliary pocket. We discovered that Numb localizes to the ciliary pocket, acting as an adapting protein to facilitate the incorporation of Ptch1 into clathrin-coated vesicles (**Figure 3**). In this process, Numb operates in a similar way to the receptor in the receptor-mediated endocytosis by connecting the “cargo” Ptch1 with adaptins. These findings contribute to a broader understanding of modulatory mechanisms governing ciliary levels of receptors and signaling transducers. We suggest that its significance is beyond the regulation of Hh signaling.

Numb participates in Ptch1 internalization at the axonal growth cone³ (Ferent et al. 2019, *published from the lab of one of the senior authors of this manuscript*). However, it's crucial to note that distinct mechanisms and molecules come into play. In the growth cone, Numb-mediated endocytosis is dependent on Boc. But Boc is absent from the ciliary pocket or any other cilium-related structure (See Author response - Figure 1, for reviewer #2 - Q3). This absence indicates that Boc is not implicated in endocytosis at the ciliary pocket. It is noteworthy that cilia often employ unique signaling transduction mechanisms due to their specialized structural settings.

- 2) We pinpointed the molecular mechanism that governs Ptch1 exit from the cilium, unraveling a long-standing mystery in Hh signal transduction. The departure of Ptch1 from the cilium is essential for the full activation of Hh signaling⁴. Results from multiple labs suggest that when Ptch1 clearance from the cilium is hindered by various factors such as Ptch1 mutations and domain truncations, the maximal activation of Hh signaling is compromised^{5,6}. However, the precise molecular details of Ptch1 departure from the cilium remains unclear. Our current study not only pinpointed Numb as an endocytic adaptor for Ptch1, but also highlighted that Ptch1 endocytosis takes place at the ciliary pocket (**Figure 3-4**).

Additionally, our findings unveiled the distinct impact of this process on the transcription factor Gli2 and Gli3 (**Figure 5-6**). We showed that when Shh-Ptch1 persists in the cilium, the ensuing Hh signaling is sufficient to abolish Gli3R production, but not strong enough to induce Gli2 activation. This results in moderate levels of Hh-target gene transcription in the nucleus, but blocks the maximal levels of Hh signaling. Importantly, the attenuated Hh signaling after Numb-

loss in NIH3T3 cells are corroborated by our results in spinal cord NPCs and GCPs in the developing cerebellum.

In a previous study, Yue et al. showed that Ptch1 levels at the cell surface were regulated by Smurf-mediated ubiquitination and the ensuing degradation in the endosome⁵. However, no direct evidence in that study showed that Smurf mediates Ptch1 endocytosis from the cilium. Although Smurf knockdown increased Ptch1 levels in the cilium, this could be an indirect consequence of overall elevated Ptch1 levels at the cytoplasmic membrane. Nevertheless, consistent with our observations in Numb-null cells, the persistence of Ptch1 in the cilium in Smurf-knockdown cells led to attenuated Hh signaling upon Shh stimulation.

- 3) Our finding that Numb is a positive regulator of Hh signaling contrasts with a previous report showing that Numb negatively regulates Hh signaling by facilitating the degradation of Gli1⁷. In the Discussion section, we present a thorough analysis to explain why the results of the 2006 study do not align with the fundamental roles of Numb in Hh signal transduction (under the subtitle “**Numb positively regulates Hh signaling**”). We believe that it is imperative to clarify the physiological role of Numb as a positive regulator of Hh signal transduction described in our current study.

Most importantly, there is very little in vivo evidence that numb serves as a general positive regulator of Hh signaling in embryonic tissues that are classically patterned by Hh ligands (e.g. the neural tube, limb, bone etc). The authors present data from the cerebellum, a tissue where HH ligands drive granule cell proliferation, but in my view they have not conclusively shown that the effects are due to HH signaling (as opposed to other effects of Numb/Numbl). Importantly Numb and Numbl mice have been analyzed extensively in the past (see for example PMID: 12410312). The CNS in these mice have no defects in temporal and spatial patterning (as would be expected of HH defects), but rather in progenitor cell maintenance. To show that Numb/Numbl are involved in Hh signaling in vivo, the authors should test the effects of their loss in other Hh-patterned tissues such as the neural tube or limb. This analysis should include both direct measurements of target gene expression and assessment of HH-driven patterning and cell differentiation effects.

We appreciate the reviewer’s suggestions on testing Numb-Hh signaling in other tissues. As this would entail the generation of novel conditional mouse knockout lines for which we do not have the appropriate Cre drivers at hand, this would involve a substantial time to perform the necessary mouse crosses to obtain the required homozygous mice. We thus leveraged the well-established system of induced differentiation of spinal cord NPCs to demonstrate Numb’s role in the developing spinal cord^{8,9}. In this approach, we generated Numb CRISPR/Cas9 knockout in mouse embryonic stem cells, and induced mESC differentiation via high Shh concentration (**new Figure 7 and Supplementary Figure 13**). We found that:

- 1) Numb loss significantly attenuated Hh signaling magnitude in NPCs.
- 2) The expression of the transcription factor that relies on high Hh signaling, Nkx2.2, was markedly reduced in Numb-KO cells; whereas the expression of transcription factors reliant on medium to low Hh signaling, Olig2 and Nkx6.1, remains moderately impacted.
- 3) Results from immunofluorescence staining revealed a significant reduction in the differentiation of NPCs into Nkx2.2-positive progenitors after Numb loss, whereas the differentiation into Nkx6.1- or Olig2-positive cells remained unaffected.

These results align with our findings in NIH3T3 cells that Numb loss attenuated the plateau level of Hh signaling and has minimal impact on low levels of Hh signaling (**Figure 5b-c**). Our results are also consistent with the reported phenotype by Petersen et al. (PMID: 12410312)¹⁰ that in *Numb cKO/Numbl-like KO* spinal cord, the Olig2-positive cell fate determination was not significantly impacted.

Major comments:

Q1: Further characterization of the TurboID cell lines is needed. In Fig.S2: Cilia length, ciliary SMO and ciliary Gli2 should be measured in response to the native ligand SHH (not just the synthetic agonist SAG).

Response: We thank the reviewer for pointing this out. Based on this suggestion, we characterized the cilia length, ciliary Smo, and ciliary Gli2 in cilium-TurboID cell lines when exposed to the ligand Shh. The results demonstrate that cilium-TurboID cells respond to Shh in a manner comparable to wild-type cells. These results are shown in **Supplementary Figure 2**. Given the similarity in results between Shh and SAG, we chose to retain the data from experiments with Shh in Supplementary Figure 2 and removed the original data with SAG.

Q2: The summary graphic in Fig.3C is misleading— there is no evidence that numb specifically co-localizes with clathrin in vesicles derived from the ciliary pocket (as opposed to vesicles derived from the plasma membrane). The authors' experiments cannot distinguish between these possibilities.

Response: We appreciate the reviewer for pointing this out. Technically, distinguishing CCVs derived from the ciliary pocket versus those from the plasma membrane is challenging. However, we posit that the CCVs marked with blue triangles in this image likely originate from the ciliary pocket for the following reasons:

- 1) The displayed image is captured from a single focal plane of the Airyscan confocal microscope, specifically focusing on the bottom of the cilium;
- 2) These CCVs are still closely associated with the cilium in the image;
- 3) We did not observe such closely clustered CCVs in other subcellular regions.

Considering these factors, the chance is very low that these CCVs could originate from the cytoplasmic membrane and subsequently clustered around the cilium.

Q3: Ciliary pocket localization of Numb is based on one marker. It needs to be confirmed by EM.

Response: We appreciate the reviewer for providing this valuable suggestion. We performed correlative light and electron microscopy (CLEM) in NIH3T3 cells expressing Numb-HA. We used nanogold-conjugated secondary antibody to improve antibody penetration. Our results show that the immunogold particles concentrate to the ciliary pocket. The results of ImmunoEM are incorporated in the revised **Figure 3c**.

Q4: In Figure 3f the co-localization results do not show that PTCH1 is internalized into Numb-containing vesicles. There is some co-localization, but also plenty of PTCH puncta that do not co-localize with numb.

Response: We thank the reviewer for pointing this out. Expansion microscopy is not the optimal technique to quantify this co-localization. We therefore accessed Numb-Ptch1 colocalization in un-expanded cells. In this study, we co-transfected cells with Numb-V5 and either Ptch1-YFP or YFP, and treated them with Shh for 30 min to induce Ptch1 exit from the cilia. We quantified the colocalization between Numb and Ptch1 in 30 cells. The results revealed a baseline co-localization between Numb and Ptch1, which was markedly increased by Shh stimulation. In contrast, no significant overlap was observed in cells co-transfected with Numb and YFP, suggesting that this co-localization is Ptch1 specific. The new results are incorporated in the revised **Figure 3h** and **supplementary Figure 6**. To prevent confusion, we have taken off the original data on images of expansion microscopy.

In the new images, there are also Ptch1-positive puncta that do not co-localize with Numb, and these Ptch1-only puncta are more evident before Shh treatment. It is beyond the scope of our current study to pinpoint the identify of these Ptch1-only puncta. However, we speculate that these puncta may represent either newly synthesized Ptch1 proteins awaiting delivery to the cilium or endocytosed Ptch1 proteins in transit to the next cellular compartment (such as the early endosome or recycling endosome).

Q5: Does endogenous numb localize in cilia, both in cells and in embryonic tissues where Hh signaling is active (ventral neural tube, limb etc.)? I understand this is technically challenging, but I believe critical to the paper.

Response: We thank the reviewer for understanding the difficulty of detecting endogenous ciliary proteins. The endogenous Numb in the ciliary pocket is not detectable in NIH3T3 cells or in embryonic tissues by immunostaining. This is likely due to its low levels in the ciliary pocket. Given its role as an endocytic adaptor protein, it is conceivable that Numb's presence in the ciliary pocket is transient, resulting in low levels that fall below the detection threshold of the Numb antibody. We would like to remind the reviewer that, in this regard, Numb is not an exception. A similar phenomenon has been noted with other ciliary proteins, such as PKA. Multiple PKA subunits were detected in the cilium in mass spectrometry studies. Further, when PKA subunits were fused to EGFP, their presence was observed in the cilium. However, efforts to detect endogenous PKA in the cilium in wild-type cells using immunostaining methods have not yielded successful results.

Nevertheless, we endeavored to establish Numb localization in the cilia of cells other than NIH3T3. We infected primary cultured granule cell precursors (GCPs) with lentiviruses expressing either Numb-GFP or Numb-V5, and co-stained with cilium marker Arl13b. Our results show that Numb localizes to the bottom section of the cilium in GCPs, in addition to its punctate localization in the cytosol. This pattern resembles the localization of Numb observed in NIH3T3 cells. We have included this result in **Figure 8a**.

Q6: PTCH1 is cleared from the cilia upon SHH addition. What is the mechanism by which SHH addition leads to Numb-mediated PTCH1 internalization? Is numb a constitutively acting on PTCH1 or only in response to SHH?

Response: We appreciate the reviewer for bringing up this insightful point. Our analysis of Numb-Ptch1 colocalization (**Fig. 3h-i**) indicates a baseline endocytosis of Ptch1 from the cilium, indicating the presence of a baseline acting on Ptch1. However, the endocytosis is notably enhanced by Shh treatment, suggesting that it is a regulated process by Shh. While the precise mechanisms governing Shh-induced Ptch1 internalization present intriguing avenues for future exploration, we speculate that upon binding to Shh, Ptch1 may adopt a conformation that promotes its interaction with Numb. This

enhanced interaction could facilitate the efficient incorporation of Ptch1 into CCVs, thereby promoting its internalization from the ciliary pocket. We have incorporated these discussion points in the Discussion section of the revised manuscript (under the subtitle “**Numb regulates protein levels in the cilia via mediating endocytosis at the ciliary pocket**”).

Q7: The authors call numb an “adaptor” for PTCH1 internalization, yet very few experiments (other than PTCH interaction) show this in a rigorous way. An adaptor links a receptor cargo to the clathrin (or other endocytic) machinery. For example, where does Numb bind to PTCH? Identifying the protein-protein interface that mediates the PTCH1-Numb interaction is critical here. The authors can then make point mutations (in both PTCH1 and Numb) that disrupt the interaction and show that there is an effect on PTCH1 clearance from cilia and Hh signaling. This type of analysis would markedly improve the paper.

Response: We thank the reviewer for raising this insightful point and apologize for any confusion in terminology. In this study, we designate Numb as an “adaptor” for Ptch1 internalization due to its interaction with Ptch1 (in this study) and adaptins (in previous studies)^{11, 12}. Numb does not directly interact with clathrin. In this context, Numb operates in a similar way to the receptor in receptor-mediated endocytosis. The cargo in this scenario is Ptch1, as opposed to the typical ligand associated with the receptor. In the revised manuscript, we used schematics to illustrate this point (**Figure 6e**), and have added such description in the related section (line #323-325, #378-379, #533-538).

In response to the reviewer’s suggestions, we further investigated the Ptch1-Numb interaction. We generated Numb truncated variants and assessed their interaction with Ptch1 via Co-IP assays (**new Supplementary Figure 5**). The results reveal that deletion of either the N-terminus (aa1-25) or the PTB domain (aa26-173) abolishes the interaction with Ptch1. Meanwhile, the N+PTB domain (aa1-173) alone show interaction with Ptch1. These results indicate that the N+PTB domain mediates Numb’s interaction with Ptch1. Notably, previous studies show that Numb interacts with adaptin and other clathrin-mediate endocytic machinery through Numb’s C-terminus¹¹⁻¹³.

We then proceed to determine which domain in Numb is essential for Hh signaling (**new Supplementary Figure 9d**). We expressed the truncated Numb variants in cells acutely depleted of Numb, and found that only full length Numb is able to rescue Hh signaling, whereas none of truncated Numb variants can. Collectively, these results suggest that both the Ptch1-interacting domains and the adaptin-interacting domains are necessary for Numb’s role in Hh signaling. These results underscore the new molecular role of Numb revealed in this study that Numb acts as an adaptor to link Ptch1 and the clathrin-mediated endocytic machinery, thereby facilitating Ptch1 exit from the cilium.

To clarify the molecular role of Numb at the ciliary pocket, we provided a schematic illustration in **Figure 6e**. This illustration emphasizes how Numb interacts with Ptch1 via one terminus and engages with the clathrin-mediated endocytic machinery at the other terminus, thereby facilitating Ptch1 endocytosis from the ciliary pocket. With these new results, Numb’s roles as an endocytic adaptor protein for Ptch1 is significantly enhanced. We therefore are grateful for the reviewer’s constructive suggestions.

Q8: The effects of Numb loss on HH signaling are partial. In fact, at lower concentration of SHH the effects are minor (e.g. Figure 5b). This should be made clear and discussed, with the explicit statement that these effects are modest.

Response: We thank the review for pointing this out. Indeed, Numb loss has minimal impact on Hh signaling at low concentration of Shh, but significantly reduces the plateau levels of Hh signaling at high

concentration of Shh stimulation, supported by results in Figure 5b. This important point was explicitly addressed in the corresponding “Result” section (line #305, line #311-314). We have also added further discussion on this point in the “Discussion” section (under the subtitle “**Numb is required for maximal activation of Hh signaling**”).

Importantly, our new results further support this critical point. We found that in neural progenitors derived from the developing spinal cord, Numb loss blocks the expression of the transcription factor that relies on high Hh signaling (Nkx2.2), but has moderate or no effect on the transcription factors reliant on intermediate-to-low Hh signaling (Nkx6.1, Olig2). We have added these results in new **Figure 7** and **Supplementary Figure 13**.

Furthermore, we generated new results that could provide mechanistic insights into the attenuated plateau levels of Hh signaling in the **new Figure 6**. In this experiment, we focused on the transcription factors Gli2 and Gli3 that control Hh signaling levels. Low levels of Shh stimulation terminates the production of Gli3R, lifting the inhibition on Hh signaling; whereas high levels of Shh stimulation activates Gli2, the primary transcriptional activator, resulting in maximal activation of Hh signaling. We found that in Numb-null cells, Shh stimulation is able to cease the proteolytic production of Gli3R, the primary transcription suppressor. However, the cilium translocation of Gli2, a critical step for Gli2 activation, is blocked. These results suggest that without Numb, cells retain the capacity to block the generation of the repressive factor Gli3R but have deficiencies in activating Gli2.

Finally, we added an analysis on the potential molecular scenario in the cilium of Numb-null cells. The persistence of Ptch1 in the cilium may lead to a ciliary environment non-conducive for optimal activation of Smo. Subsequently, the downstream signaling of Smo is only sufficient to cease Gli3R production but not robust enough to enable Gli2 activation. We included this analysis in the “Discussion” section (line #600-617).

Q9: In Figure 4, significant additional work needs to be done to establish that Numb loss impairs PTCH removal from cilia in response to SHH. First, this needs to be done with endogenous PTCH1, not PTCH1 artificially (non-physiologically) elevated after SAG addition.

Response: We appreciate the reviewer’s constructive suggestions. We agree that it is ideal to quantify the endogenous levels of Ptch1 in the cilium. However, it's worth noting that the levels of Ptch1 in the cilium at the resting stage fall below the detection limit of immunostaining. We are not aware of any literature documenting the detection of endogenous Ptch1 in the cilium in wild-type cells prior to Hh pathway activation. The methodology employed in our study, SAG-induced Ptch1 expression and enrichment in the cilium followed by Shh-triggered Ptch1 exit from the cilium, was initially described in a study from Matthew Scott’s lab⁴. This is the original study that describes Ptch1 departure from the cilium upon Shh stimulation. This approach is widely accepted as a reliable method for investigating Ptch1 trafficking during Hh activation.

However, to validate this result, we performed additional experiments with an alternative approach. We infected wild-type and Numb KO cells with lentiviruses expressing Ptch-YFP to ensure low and uniform expression levels across cells (**new Supplementary Figure 8a**). Under this condition, Ptch1-YFP is observed in the cilia prior to Shh stimulation. We then incubated cells with Shh and quantified the ciliary Ptch1 levels after Shh treatment. Our results show that Shh significantly reduced ciliary Ptch1-YFP

levels in wild-type cells. In contrast, Ptch1-YFP levels remained unchanged before and after Shh in Numb KO cells.

Taken together, we used two different approaches to demonstrate that Numb loss impairs Ptch1 removal from the cilium in response to Shh stimulation. We trust the reviewer will concur that these results offer strong evidence to substantiate our statement.

- *Second, the protein abundance of PTCH1 in cilia should be clearly plotted - and + SHH in both WT and numb ko cells. Plotting a ratio (as the authors have done) is not appropriate since the ratio is generated using a mean value for baseline (and is therefore not a single cilium measurement).*

Response: We appreciate the reviewer's suggestion. In the revised figures, we plotted the raw data points of Ptch1 protein intensity in individual cilium before and after Shh treatment (**Figure 4c, new Supplementary Figure 8b**).

Q10: In Figure 5, SmoM2 overexpression is used to show that the requirement of Numb is at the level of PTCH1, not SMO. SmoM2 over expression is quite non-physiological, so the authors should test whether Numb inactivation influences signaling by endogenous SMO in response to agonists like SAG and to genetic ablation of negative regulators of signaling downstream of SMO (e.g. GNAS, SUFU).

Response: We appreciate the reviewer's suggestions and conducted the following experiments.

- 1) We simulated wild-type (WT) or Numb KO cells with SAG and assessed Hh signaling activity by measuring Gli1 transcript levels. The results show that at all concentration of SAG, the Hh signaling levels are comparable between WT and Numb KO cells (**new Figure 5g**). Note that Hh signaling activity reaches plateau levels in WT cells at around 60nM SAG, and the plateau levels are comparable in WT and Numb KO cells.
- 2) We knocked down Sufu in WT and Numb KO cells. We found that Sufu knockdown turned on Hh signaling independent of Shh in both WT and Numb KO cells (**new Supplementary Fig. 10b, c**). This result suggests that Numb acts upstream of Sufu.
- 3) We expressed in WT or Numb-KO cells the cilium-targeting PKI, a PKA peptide inhibitor tagged to Arl13b-N-RVEP-PR. The ciliary PKI was used in a few previous studies to specifically inhibit PKA activity in the cilium^{14, 15}. Our results show that cilium-PKI triggered Hh signaling to the similar magnitude in WT and Numb KO cells (**new Supplementary Fig. 10d**).

Taken together, these results further reinforce the notion that Numb operates upstream of Smo to regulate Hh signaling.

Reviewer #2:

In this work, Liu et al. identified Numb as a novel ciliary protein from using an impressive cilia membrane-specific proximity labeling approach. Using cells in culture they found Numb localizes at the ciliary pocket and clathrin-associated vesicles. Experiments in 3T3 cells revealed that similar to the previously reported function of Numb in the non-canonical Shh pathway, Numb mediates the removal of Ptch1 from primary cilia and also that it augments activation of the canonical Shh pathway. They show that knockout or knockdown of Numb in cells partially blocks Ptch1 removal from cilia and compromises Shh

signaling. In mice, conditional knockout of Numb along with a null allele of Numb1 in GCPs results in a mild reduction in HH signaling in GCPs and it is claimed to reduce proliferation of GCPs, and it also causes a smaller cerebellum during development and in the adult. The quality of the data are solid, and most conclusions are appropriate. The findings of this work add to the known roles of Numb in Ptch1 removal by showing a function in the canonical Shh pathway. Overall, the paper will be a valuable Resource paper for cilia and SHH specialists and it also presents a standard phenotype analysis in the cerebellum of one protein the authors chose to follow up on (Numb). The paper has several limitations for a high impact paper.

We appreciate the reviewer for pointing out that our cilia membrane-specific proteomics is impressive. We are thankful for the recognition that our data is solid, and the conclusions are appropriate. We are also grateful for the constructive suggestions provided by the reviewer. In response, we conducted several additional experiments and provided the following point-by-point answers. We believe these additions have considerably enhanced the significance of our studies.

Major concerns:

(1) *The new findings seem incremental given that two previous papers showed endocytosis is involved in Ptch1 removal from cells. One paper showed that Ptch1 removal from primary cilia is mediated by a Smurf-related endocytosis and subsequent lysosome degradation and this leads to reduced GCP proliferation, and another paper showed that Ptch1 removal from the cell surface in non-canonical SHH signaling involves Numb and endocytosis dependent on Boc.*

Response: We thank the reviewer for pointing out the related publications that align with our current study. We kindly refer to our response to Reviewer #1 for the novelty and significance of our current study, and a more detailed response. However, we want to highlight two key points:

- A) In a previous study, Yue et al. showed that Ptch1 levels at the cell surface were regulated by Smurf-mediated ubiquitination and the ensuing degradation in the endosome⁵. However, no direct evidence in that study showed that Smurf mediates Ptch1 endocytosis from the cilium. Although Smurf knockdown increased Ptch1 levels in the cilium, this could be an indirect consequence of overall elevated Ptch1 levels at the cytoplasmic membrane. Nevertheless, consistent with our observations in Numb-null cells, the persistence of Ptch1 in the cilium in Smurf-knockdown cells led to attenuated Hh signaling upon Shh stimulation.
- B) Numb participates in Ptch1 internalization from the axonal growth cone³ (Ferent et al. 2019, *published from the lab of one of the senior authors of this manuscript*). However, it's crucial to note that distinct mechanisms and molecules come into play. In the growth cone, Numb-mediated endocytosis is dependent on Boc. But Boc is absent from the ciliary pocket or any other cilium-related structure (See Author response - Figure 1, for reviewer #2 - Q3). This absence indicates that Boc is not implicated in endocytosis at the ciliary pocket. It is noteworthy that cilia often employ unique signaling transduction mechanisms due to their specialized structural settings.

(2) *The Shh/Ptch1-related phenotypes (removal of Ptch1 and GCP expansion) of Numb mutants appear mild, raising the question of whether Numb plays only a minor role in removing Ptch1. Furthermore, the mechanism demonstrated in vitro was not validated in a second relevant cell type, such as a human RPE cell line.*

Response: We thank the reviewer for pointing this out. Regarding the impact of Numb loss on Hh signaling activity, we kindly refer to our answer to reviewer #1 - Q8. Regarding the impact of Numb loss on Ptch1 removal phenotype, we now plotted the raw data (instead of the ratio in our original manuscript) on Ptch1 intensity in the cilium (**Figure 4c, new Supplementary Figure 8b**). In WT cells, the ciliary Ptch1 intensity significantly decreased after Shh stimulation. In contrast, in Numb KO cells, the ciliary Ptch1 intensity remained unchanged before and after Shh stimulation.

To demonstrate the effect of Numb KO on cell types beyond NIH3T3, we employed CRISPR/Cas9 knockout in spinal cord neural progenitor cell (NPCs) (**new Figure 7**), conducted Numb knockdown in primary cultured GCPs (**new Supplementary Figure 14**), and established Numb genetic knockout in GCPs (**Figure 8**). We do agree that the effect on the cerebellum appears mild; however, it is a significant and reproducible effect. In addition, it is not rare that in an *in vivo* context (such as the cerebellum), the observed effect appears milder than in an *in vitro* isolated system (such as a cell line). This might be due to the robustness of the *in vivo* situation by mechanisms such as compensation, or it might be due to technical aspects such as the time that it takes for the Cre to act on the Numb floxed gene and lead to protein decrease. In addition, it has been shown that upon postnatal cerebellum insults, other progenitor cells can partially replace injured GCPs in the generation of the cerebellum and compensate for the injury¹⁶. While investigating this aspect is beyond the goal of our study, it nevertheless points to the resilience that the cerebellum has in an *in vivo* context. Nonetheless, the findings across these various cell types (NPCs and GCPs) align with our observations in NIH3T3 cells, demonstrating that Numb KO reduces the maximal levels of Hh signaling while leaving the lower levels of Hh signaling unaffected.

(3) *It is not clear if the function of Numb on Ptch1 removal from cilia relies on Boc (as in non-canonical signaling) or not, notably Boc is not listed in the ciliary proteome and Izzi et al. showed Boc mutants have a GCP proliferation phenotype. This point needs to be addressed.*

Response: We thank the reviewer for making this valuable suggestion. To determine whether Boc is also involved in Ptch1 internalization in the ciliary pocket, we expressed Boc-YFP in NIH3T3 cells. We found that Boc does not exhibit localization to the cilium or any adjacent structures (**Author response - Figure 1**). Further, Shh stimulation does not induce any noticeable alterations in Boc's subcellular localization. Therefore, Boc is not at the right place to facilitate Ptch1 exit from the cilium. We included this result below as Author response - Figure 1.

Author response - Figure 1: Boc-YFP was expressed in NIH3T3 cells and co-immunostained with the cilium marker Arl13b. Boc does not exhibit localization to the subcellular structure associated with the primary cilium, such as ciliary axoneme, basal body, or ciliary pocket.

(4) *The authors should address whether Numb plays a role in cilia formation in GCPs in vivo or maintaining the ciliary pocket compartment in vivo or in vitro.*

Response: We thank the reviewer for making this important suggestion. To assess the maintenance of the ciliary pocket in Numb KO cells, we labeled the ciliary pocket with EHD1-GFP and performed expansion microscopy to visualize the ciliary pocket. The results show that there are no discernible differences in the morphology of ciliary pocket between the WT and Numb KO cells (**new Supplementary Figure 8c**).

We also assessed Numb KO's effect on cilium formation in GCPs. Considering the technical challenges in examining cilia morphology in GCPs in vivo, we knocked down Numb expression with lentiviruses in primary culture GCPs and labeled the cilium with Arl13b. We found no discernible changes in the cilium formation after Numb knockdown, including the cilium length and the Smo localization to the cilium (**new Supplementary Figure 14c, f**).

(5) Does Numb control the protein levels of other SHH signaling-related membrane proteins in GCPs, e.g. is there decreased Boc, Smo accumulation in cilia of mutants after stimulation of SHH signaling?

Response: We appreciate the reviewer for the important suggestions. As illustrated in our answer to Q3, Boc does not localize to the cilium or cilium related subcellular sites, and Shh treatment does not impact its subcellular localization.

To determine Smo accumulation in the cilium, we conducted Smo immunofluorescence staining in NIH3T3 cells (**new Supplementary Figure 11**) and in primary cultured GCPs (**new Supplementary Figure 14 f-g**). The results show that Numb KO does not impact Shh-induced Smo intensity in the cilium in both cell types. Under some circumstance, Smo may accumulate in the cilium but assume inactivation conformation, as documented by Rohatgi et al., 2009¹⁷. We think that in Numb KO cells, either the ciliary Smo molecules adopt an inactive conformation similar to that induced by cyclopamine, or only a partial population of Smo is activated. Under either condition, the downstream signaling is activated partially to cease Gli3R production but not robustly enough to enable Gli2 activation. We included a detailed discussion on this point under the subtitle "**Numb is required for maximal activation of Hh signaling**" in the Discussion section.

(6) The phenotype of Math1-Cre; Numb-fl/fl mice must be presented to determine whether the loss of Numbl contributes to the mild phenotype reported in DCKO.

Response: We thank the reviewer for this comment. We would like to summarize the multiple lines of evidence suggesting that Numbl-like is not essential for the regulatory mechanism of Hh signaling described in our current study.

First, unlike Numb, Numbl-like does not exhibit as discrete puncta in the cytosol (**Supplementary Figure 12a**), suggesting that it may not involve in clathrin-mediated endocytosis. Second, Numbl-like knockdown does not impact Hh signaling or exacerbate the reduced Hh signaling in Numb KO cells (**Supplementary Figure 12b-c**). Third, in primary cultured GCPs from Numbl-like-null mice, Shh markedly induce Hh signaling (comparing -Shh and +Shh condition in **Figure 8b**). In contrast, removing Numb in Numbl-null background significantly reduced Hh signaling (comparing the +Shh conditions in **Figure 8b**). Finally, we examined the effect of removing Numbl-like alone on cerebellar development. The results show that genetic removal of Numbl-like has no significant impact on the P6 cerebellar size (**Supplementary Figure 15b**). In summary, our findings in NIH3T3 cells and results from the mouse model all point in the same direction, suggesting that Numb, rather than Numbl, is important for Hh signaling; the observed GCP proliferation phenotype is primarily attributed to the loss of Numb.

(7) *Numb* expression appears to be much higher in Purkinje cells compared to GCPs, but Purkinje cells do not transduce HH signaling. What do the authors think is the function of the high level of *Numb* in Purkinje cells?

Response: We appreciate the reviewer for pointing this out. We are also intrigued by the high expression of *Numb* in Purkinje neurons. However, this interesting question is beyond the scope of our current manuscript.

(8) *In the mouse experiments, state the genotype of controls. Are they Numbl nulls? If not, they should be.*

Response: All genotypes of control mice are now listed in the figure legend of Figure 8. In Supplementary Fig. 15b, we show that inactivating *Numbl* has no impact on the cerebellum area, in line with *Numb*, but not *Numbl*, playing an important role in the Hh pathway.

(9) *Fig. 7 The number of pH3+ cells per lobe cannot be used as a measure of proliferation rate since the DCKO have a smaller cerebellum. The measurement should be PH3+Ki67+/Ki67+ cells or PH3+ cells per same area of outer EGL, and the same lobule and medial-lateral position used.*

Response: We thank the reviewer for the suggestions. We confirm that the quantification was performed at the same lobule and medial-lateral position. We have included the quantification results of PH3+ cells per 600 μ m of outer EGL (**Figure 8i**).

(10) *“The length of cell cycle was calculated by: 2h x (PCNA+ cells/BrdU+EdU- cells).” This is not an accurate measure since most of BrdU is cleared by 1 hour. To accurately measure cell cycle length, the authors would need to inject EdU at different times from 12-24 hours after BrdU administration and determine when maximum double labeling occurs.*

Response: We appreciate the reviewer’s suggestion. In addition to the reviewer’s point, since we find that the duration of the cell cycle is not directly pertinent to the primary focus of this study, we have opted to remove this data from our manuscript.

(11) *Fig. 8a, d. Please orient cerebella so dorsal is up and anterior is consistently to left (or right). Midsagittal sections should be shown to be able to interpret the phenotype. The H&E staining in (d) needs to be improved. Furthermore, in the adult DCKO, a lobule seems to be missing in the central region (L7?). Is this a consistent phenotype?*

Response: We thank the reviewer for the suggestions. We followed the suggestions and re-oriented the cerebellar images (new **Figure 8j**). We improved the H&E staining for adult mouse brain (new **Supplementary Figure 15c**). We also labeled the cerebellar lobules. In the revised figure, we used arrowheads to indicate under-developed lobules in the cDKO cerebellum, which is a consistent phenotype.

(12) *Bergmann glia respond to SHH and thus must have cilia. Do they express Numb and does it play a role in SHH signaling? A high magnification image should be included with double labeling for Numb and a Bergmann glia marker and/or Purkinje cell marker.*

Response: We appreciate the reviewer for these suggestions. While we acknowledge that there are intriguing questions regarding Bergmann glia and Purkinje neurons, we consider that these inquiries belong to separate projects that extend beyond the scope of our present study.

(13) To examine Shh activation in the Numb mutants, besides Gli1 and Ptch1 transcription, the enrichment of Smo and GLI trafficking in cilia should be examined in the mutants, in particular, because if Ptch1 fails to move out of the cilium it has been reported this should block Smo entry into the cilium.

Response: We appreciate the reviewer's suggestions. Kindly refer to our response to Reviewer #2 - Q5 regarding Smo enrichment in the cilium.

We also examined the cilium translocation of Gli2, and we found that Shh-induced cilium transport of Gli2 was significantly reduced in Numb KO cells compared with WT cells (**Figure 6a-b**). Kindly refer to our responses to reviewer #1 - Q8 about the impact of Numb KO on Gli2 transcription factor.

Minor comments:

(1) Fig. 7 The IF staining of Numb in the EGL of the cerebellum seems very low and questionable. Higher magnification images with the negative control (Cre+) are required. Also, qRT-PCR would strengthen the results.

Response: We thank the reviewer for the suggestions. In the revised manuscript, we updated these images with new immunofluorescence staining experiments (**new Figure 8d**), and provided high magnification images. In **Figure 8f**, we provided Western blot results showing that Numb protein levels are significantly reduced in isolated GCPs from the cDKO. We believe this is stronger evidence than qRT-PCR as it shows the effect on the protein.

(2) Fig. 6C, D please make it clear what “” indicates - significant differences between groups or time points?*

Response: We apologize for the confusion. The “*” or “ns” indicates significant differences between “control” and “cDKO” at the highest concentration of Shh (30 nM) or Purmorphamine (150nM) treatment. These two graphs are now presented in new **Figure 8g**.

(3) Fig. 7. Low power images of the cerebella indicating where the high power images are taken from should be included. Also, the lobules the quantification was performed on should be stated.

Response In the revised manuscript, we added dashed boxes to indicate where the zoomed-in images are taken from low-magnitude images. The quantification was performed on lobules VIII and IX (**new Figure 8d, h**).

(4) All data points should be shown in graphs to illustrate variation more accurately than bar graphs.

Response: Thanks for the suggestion. We have now added the data points in the revised figures.

Reviewer #3:

In this manuscript, Liu et al use turbo-ID technology to elucidate the protein composition of the cilium in response to Shh treatment and discover that Numb regulates the departure of Ptch1 from the cilium and regulates cerebellar size in mice. This is a highly rigorous and fascinating glimpse into the

mechanistic basis of Ptch1 regulation of Shh signaling. I deeply appreciated the NUMB KO/reconstitution data and the SMO2 stimulation data which nicely illustrated where Numb works in the signaling pathway. The proteomic experiments are conducted with the utmost care and the methods used are highly rigorous and state-of-the-art. I only have small suggestions to increase the clarity of the proteomic parts of the paper. The Cre conditional Numb KO data impressively illustrated the in vivo importance of the biochemical findings in the NIH3T3 cells. Overall, it was a fantastic paper.

We thank the reviewer for pointing out that our proteomic experiments are conducted with “utmost care” and that our study is a “highly rigorous and fascinating glimpse into the mechanistic basis of Ptch1 regulation of Shh signaling”. We also appreciate the reviewer for acknowledging our characterization of where Numb acts in Hh signaling and our *in vivo* study with Cre-mediated conditional Numb knockout. We are delighted that the reviewer thought that our work was “a fantastic paper”.

Minor Criticism:

1) *What was the exact distribution of samples into TMT channels and if a second TMT mix was employed, how did it differ from the first? Was the pool channel only used to calculate enrichment or was it also used to expand the TMT experiment into multiple mixes? A supplemental methods figure illustrating the distribution of all samples into TMT channels would help to clarify this point.*

Response: We thank the reviewer’s suggestions. For the TMT labeling, we did not use a second TMT mix. We used a single TMTpro 16plex label kit (ThermoFisher Cat # A44521, Lot # VC294912). The pool channel was for normalization of protein intensity in a single run. The distribution of all samples into TMT channels is included in the new **supplementary Figure 3a**.

2) *Typically, high pH reversed phase offline fractionation is employed prior to SPS MS3 due to the long duty cycle for SPS MS3. Was that not done in this case? The methods mention LCMS of each "fraction" but no details of what method was used to fractionate.*

Response: We performed fractionation using a high pH reverse phase approach, with the Pierce High pH Reversed-Phase Peptide Fractionation Kit (Cat # 84868) kit. This description has been added to the Methods section under Mass Spectrometry. We thank the reviewer for pointing this out.

3) *For the volcano plots (Fig 2d, S3), how were the p values calculated and adjusted for multiple comparison?*

Response: P values were calculated from two-tailed, unpaired Student’s t tests. The P values were adjusted for multiple testing via the Benjamini-Hochberg procedure.

4) *Fig 2C could be combined with an array of replicate-to-replicate reporter intensity scatter plot in the supplement mentioning the R squared correlation of the replicate pairs on the plots and mentioning the average R squared in the text of the paper. The heatmap does not quantify the correlation quite as rigorously as that although the heatmap does look very impressive.*

Response: We appreciate the reviewer’s suggestion. As suggested, we generated the replicate-to-replicate reporter intensity scatter plots and included the R-square correlations on the plots. The results are integrated into the **new Supplementary Figure 4a-d**. We also added the averaged R-squared in the figure legend for the corresponding figure.

5) All proteomic data needs to be deposited into the PRIDE database prior to publication and the proteomic data acquisition parameters should be described in more detail in the methods or supplement.

Response: We thank the reviewer for the suggestions. The proteomic data are deposited into MassIVE, which is part of proteomeXchange. The data will be available to the public once this manuscript is accepted for publication. We added the details of proteomic data acquisition parameters in the Methods section under "Mass spectrometry".

5) For fig 5d, what was the concentration of Shh used. this information will be helpful in comparison of 5d to 5b.

Response: The Shh concentration in Figure 5d (New Figure 5e) corresponds to the highest Shh concentration in 5b.

6) possible typo in line 499; impacts seems like it should be impact.

Response: Thanks for pointing this out. It has been corrected.

Reviewer #4:

This reviewer was asked to evaluate the ExM imaging. The reviewer finds the data well presented, the method well-described and the immunostaining and results convincing. These are very nice ExM data of cilia. However, the wording is not standard for the description of the data.

Using the term "superresolution microscopy" in the way the authors do in the manuscript in line 214 line 1025, 1034 is in the reviewers opinion not useful. Yes, the expansion results in the resolution of structures that may be separated by distances below the diffraction limit and Airyscan microscopy yields somewhat of a resolution increase, but "superresolution microscopy" is not a technically accurate term. The authors performed Expansion microscopy or they expanded the sample and then they imaged using an airy scan confocal microscope yielding an apparent increase in resolution. They do describe it better later in the manuscript.

Response: We thank the reviewer for bringing this to our attention. Following the reviewer's suggestion, we have removed "superresolution microscopy" and adjusted the description accordingly in all instances.

References

1. Ghossoub, R., Molla-Herman, A., Bastin, P. & Benmerah, A. The ciliary pocket: a once-forgotten membrane domain at the base of cilia. *Biology of the Cell* **103**, 131-144 (2011).
2. Molla-Herman, A. *et al.* The ciliary pocket: An endocytic membrane domain at the base of primary and motile cilia. *Journal of Cell Science* **123**, 1785-1795 (2010).
3. Ferent, J. *et al.* Boc Acts via Numb as a Shh-Dependent Endocytic Platform for Ptch1 Internalization and Shh-Mediated Axon Guidance. *Neuron* **102**, 1157-1171.e1155 (2019).
4. Rohatgi, R., Milenkovic, L. & Scott, M.P. Patched1 regulates hedgehog signaling at the primary cilium. *Science* **317**, 372-376 (2007).
5. Yue, S. *et al.* Requirement of Smurf-mediated endocytosis of Patched1 in sonic hedgehog signal reception. *Elife* **3** (2014).
6. Tukachinsky, H., Petrov, K., Watanabe, M. & Salic, A. Mechanism of inhibition of the tumor suppressor Patched by Sonic Hedgehog. *Proc Natl Acad Sci U S A* **113**, E5866-E5875 (2016).
7. Marcotullio, L.D. *et al.* Numb is a suppressor of Hedgehog signalling and targets Gli1 for Itch-dependent ubiquitination. *Nature Cell Biology* **8**, 1415-1423 (2006).
8. Gouti, M. *et al.* In vitro generation of neuromesodermal progenitors reveals distinct roles for wnt signalling in the specification of spinal cord and paraxial mesoderm identity. *PLoS Biol* **12**, e1001937 (2014).
9. Pusapati, G.V. *et al.* G protein-coupled receptors control the sensitivity of cells to the morphogen Sonic Hedgehog. *Science Signaling* **11**, aao5749-aa05749 (2018).
10. Petersen, P.H., Zou, K., Hwang, J.K., Jan, Y.N. & Zhong, W. Progenitor cell maintenance requires numb and numblake during mouse neurogenesis. *Nature* **419**, 929-934 (2002).
11. Berdnik, D., Török, T., González-Gaitán, M. & Knoblich, J.A. The Endocytic Protein α -Adaptin Is Required for Numb-Mediated Asymmetric Cell Division in *Drosophila*. *Developmental Cell* **3**, 221-231 (2002).
12. Song, Y. & Lu, B. Interaction of Notch signaling modulator Numb with alpha-Adaptin regulates endocytosis of Notch pathway components and cell fate determination of neural stem cells. *J Biol Chem* **287**, 17716-17728 (2012).
13. Smith, C.A., Dho, S.E., Donaldson, J., Tepass, U. & McGlade, C.J. The cell fate determinant numb interacts with EHD/Rme-1 family proteins and has a role in endocytic recycling. *Mol Biol Cell* **15**, 3698-3708 (2004).
14. Mick, D.U. *et al.* Proteomics of Primary Cilia by Proximity Labeling. *Developmental Cell* **35**, 497-512 (2015).
15. Happ, J.T. *et al.* A PKA inhibitor motif within SMOOTHENED controls Hedgehog signal transduction. *Nat Struct Mol Biol* **29**, 990-999 (2022).
16. Wojcinski, A., Lawton, A.K., Bayin, N.S., Lao, Z., Stephen, D.N. & Joyner, A.L. Cerebellar granule cell replenishment postinjury by adaptive reprogramming of Nestin(+) progenitors. *Nat Neurosci* **20**, 1361-1370 (2017).
17. Rohatgi, R., Milenkovic, L., Corcoran, R.B. & Scott, M.P. Hedgehog signal transduction by Smoothened: Pharmacologic evidence for a 2-step activation process. *Proceedings of the National Academy of Sciences* **106**, 3196-3201 (2009).

REVIEWER COMMENTS

Reviewer #1 (Remarks to the Author):

In this manuscript, Liu, Yam and colleagues report that Numb localizes to the ciliary pocket and acts as a positive regulator of Hedgehog (Hh) signaling. They show Numb acts as an endocytic adaptor to incorporate Ptch1 into clathrin-coated vesicles, thereby promoting Ptch1 exit from the cilium, a key step in Hh signaling activation. Overall, the data are of high quality and strongly support the key conclusions. The proteomic dataset results from well-controlled experiments and should also be of interest to the field. That Numb is a positive regulator of Hh signaling has been reported previously. The published work also raises the question of whether Numb is a general regulator of Hh signaling or only does so in specific contexts. This question is important because the phenotypes exhibited by mouse embryos mutant for Numb (and Numbl) are very different from that exhibited by Hh mutants (i.e. many aspects of Hh signaling are not affected by Numb and Numbl loss). The current study provides a definitive answer to this question that Numb only regulates Hh signal in specific contexts and is only required for high level Hh response. Thus, the manuscript represents a clear conceptual advance to warrant publication in Nature Communications.

The authors have made great effort to address the issues raised by the reviewers, and the additional data provided significantly enhanced the manuscript. I only have two minor issues that need to be addressed before the manuscript is acceptable for publication:

(1) The conclusion that Numbl is not essential for Hh signaling likely is incorrect. The authors base this conclusion on Numbl knockdown in NIH3T3 cells, showing a lack of an effect on Hh signaling. But it is contradicted by mouse knockout experiments. In the latter, knocking out both Numb and Numbl is required for affecting cerebellum development, whereas knocking out either Numb or Numbl alone has no effect. In other words, Numb and Numbl are redundant in this context. This raises the question of whether the effect seen in mice is Hh-independent or, more likely, the lack of effect of Numbl loss is due to its absence in NIH3T3 cells. Previous studies have shown that Numbl is not as widely expressed as Numb (i.e. there are tissues when Numbl is not expressed).

(2) The title does not reflect the significance of the findings. That Numb is a positive regulator of Hh signaling has been reported previously, and cilium proteomics is not directly connected to the novel mechanistic insights into how Numb acts with regard to Hh signaling.

Reviewer #2 (Remarks to the Author):

In this revised version, the authors provided further experiments and data to illustrate (1) defined the interaction domain of Numb with Ptch1; (2) Numb controls only maximum activation levels of HH

signaling (GLI2A) but not low levels of HH signaling (GLI3R); (3) Numb functions in mediating HH signaling-dependent ESC differentiation into NPCs as an in vitro model for neural tube patterning (see below); (4) localization of Numb to the base of cilia in GCPs in culture to confirm previous data in NIH 3T3 cells. These additional data provide more evidence for the conclusions of the work.

The novelty of this study however is still a major concern. Below is a point by point response to the issues from the previous review that were not addressed and some new concerns:

Original major concerns remaining:

(1) The new findings seem only incremental: 1) The data indicates that Numb does not mediate the dominant machinery for removal of Ptch1 from cilia, especially since Numb ko/kd or Smurf-kd compromise Ptch1 removal from cilia and as a consequence compromise HH signaling activation. 2) A large number of membrane proteins were shown to be removed from cilia upon SHH stimulation (comparing membrane proteins identified from + SHH + biotin verse -SHH + biotin group), however the impact of this interesting finding was not addressed. Further discussion/investigation on the differences among these two groups of proteins and whether other HH signaling-related membrane proteins are impacted in Numb mutant cells, either by immune-staining or proximal labeling, would strengthen the paper.

(2) The new analyses of Numb ko in ESC-derived NPCs and GCPs in culture strengthen the mechanistic aspect of part by providing some relevance for normal cell types in their in vivo setting. Analysis of in vivo cells (neural tube and GCPs or limb mesenchyme) as previously requested would greatly strengthen the study. Furthermore, if Numb is indeed efficiently removed in conditional ko mutants from GCPs at P6 (in Figure 8 d, f), then the mild cerebellar phenotype indicates a Numb function with other proteins in activating SHH signaling.

(5) The result showing SMO is present in Numb ko cells is convincing. The presence of SMO and PTCH1 in Numb mutant cilia is controversial but intriguing.

(11) The lobule information should be labeled in the image and/or listed in the legend of Figure 8 d and h. Supplementary Figure 15c H&E control image the tissue is broken, high quality data should be shown.

Questions from newly added data:

(1) Since Numb acts at the cilium pocket and as an endocytic adaptor to remove Ptch1, in cell types such as ESCs and MEFs that lack a cilium pocket would this role be diminished or replaced by other factors? This should be discussed.

(2) Since for technical reasons the authors cannot detect endogenous Numb with a Numb antibody, they used tagged NUMB protein to stain ciliary compartments. The authors should provide quantification of the % of cells that have NUMB (Numb-V5 or -GFP) localized to the cilia pocket/base of total cells with tagged protein. This is important because in Figure 3h and supplemental Figure 6a, the Numb-V5 channel shows no cilia base staining. Also to address is the Numb localization to the cilium pocket

related to the level of SHH signaling?

Reviewer #3 (Remarks to the Author):

Thank you for addressing all of my minor criticisms of the original submission. All of my concerns are addressed now.

Reviewer #4 (Remarks to the Author):

the authors addressed my concerns fully.

Responses to Reviewers

Dear reviewers,

Thank you very much for reviewing our revised manuscript. We appreciate the positive feedback and additional revision suggestions from the reviewers. Based on these suggestions, we have performed new experiments and quantifications, and made corresponding changes in the figures and text. We believe that these revisions further strengthened our manuscript. Please find a point-by-point response to reviewers' comments below, with reviews marked in *italics* and our responses in blue text. We have highlighted the changes in the manuscript in blue text.

Reviewer #1:

In this manuscript, Liu, Yam and colleagues report that Numb localizes to the ciliary pocket and acts as a positive regulator of Hedgehog (Hh) signaling. They show Numb acts as an endocytic adaptor to incorporate Ptch1 into clathrin-coated vesicles, thereby promoting Ptch1 exit from the cilium, a key step in Hh signaling activation. Overall, the data are of high quality and strongly support the key conclusions. The proteomic dataset results from well-controlled experiments and should also be of interest to the field. That Numb is a positive regulator of Hh signaling has been reported previously. The published work also raises the question of whether Numb is a general regulator of Hh signaling or only does so in specific contexts. This question is important because the phenotypes exhibited by mouse embryos mutant for Numb (and Numbl) are very different from that exhibited by Hh mutants (i.e. many aspects of Hh signaling are not affected by Numb and Numbl loss). The current study provides a definitive answer to this question that Numb only regulates Hh signal in specific contexts and is only required for high level Hh response. Thus, the manuscript represents a clear conceptual advance to warrant publication in Nature Communications.

The authors have made great effort to address the issues raised by the reviewers, and the additional data provided significantly enhanced the manuscript. I only have two minor issues that need to be addressed before the manuscript is acceptable for publication.

We are grateful to the reviewer for acknowledging that our data is “high quality and strongly support the key conclusion”. We appreciate the recognition that our study “represents a clear conceptual advance” in the field, and the endorsement of its publication in Nature Communications. Please find our detailed responses to the reviewer's comments below.

(1) The conclusion that Numbl is not essential for Hh signaling likely is incorrect. The authors base this conclusion on Numbl knockdown in NIH3T3 cells, showing a lack of an effect on Hh signaling. But it is contradicted by mouse knockout experiments. In the latter, knocking out both Numb and Numbl is required for affecting cerebellum development, whereas knocking out either Numb or Numbl alone has no effect. In other words, Numb and Numbl are redundant in this context. This raises the question of whether the effect seen in mice is Hh-independent or, more likely, the lack of effect of Numbl loss is due to its absence in NIH3T3 cells. Previous studies have shown that Numbl is not as widely expressed as Numb (i.e. there are tissues when Numbl is not expressed).

Response: We thank the reviewer for the insightful comments. We agree with the suggestion that the lack of effect of Numb1-loss in NIH3T3 cells could be attributed to its absence in this cell line. However, this is unlikely as the expression of Numb1 in NIH3T3 cells has been confirmed by Western blot analysis in a previous study¹. Further, our qPCR results suggest that Numb1 mRNA is readily detectable in NIH3T3 cells (Supplementary Fig. 12). Nonetheless, while our data suggest that Numb1 is less likely to be involved in the regulation of Ptch1 exit from the cilium, we cannot entirely rule out its role in this process, as the reviewer suggests. Thus, we have changed the text accordingly (Line # 376-377 and 484-485).

(2) The title does not reflect the significance of the findings. That Numb is a positive regulator of Hh signaling has been reported previously, and cilium proteomics is not directly connected to the novel mechanistic insights into how Numb acts with regard to Hh signaling.

Response: We thank the reviewer for the valuable suggestion on the title. We have changed the title to **“Numb positively regulates Hedgehog signaling at the ciliary pocket”**.

Reviewer #2:

In this revised version, the authors provided further experiments and data to illustrate (1) defined the interaction domain of Numb with Ptch1; (2) Numb controls only maximum activation levels of HH signaling (GLI2A) but not low levels of HH signaling (GLI3R); (3) Numb functions in mediating HH signaling-dependent ESC differentiation into NPCs as an in vitro model for neural tube patterning (see below); (4) localization of Numb to the base of cilia in GCPs in culture to confirm previous data in NIH 3T3 cells. These additional data provide more evidence for the conclusions of the work.

We thank the reviewer for acknowledging that our additional data “provide more evidence for the conclusions”. To address the remaining concerns of this reviewer, we provide the following point-by-point responses.

The novelty of this study however is still a major concern. Below is a point by point response to the issues from the previous review that were not addressed and some new concerns.

Original major concerns remaining:

(1) The new findings seem only incremental: 1) The data indicates that Numb does not mediate the dominant machinery for removal of Ptch1 from cilia, especially since Numb ko/kd or Smurf-kd compromise Ptch1 removal from cilia and as a consequence compromise HH signaling activation.

Response: We appreciate the review’s question regarding the molecular machinery that mediates Ptch1 exit from the cilium. We apologize for not explicitly clarifying this question in our previous response and would like to address the reviewer’s concern as follows.

Our results suggest that Numb, functioning as an endocytic facilitator, plays a direct role in facilitating Ptch1 endocytosis at the ciliary pocket. This is supported by Numb’s localization to the ciliary pocket (Figure 3), its interaction with both Ptch1 (Figure 3, Supplementary Figure 5) and adaptins, and its impact on ciliary Ptch1 levels and Hh signaling (Figure 4 - 6).

In contrast, Smurf proteins are E3 ubiquitin ligases that add ubiquitin to target proteins. Smurfs do not directly partake in protein endocytosis; instead, they mediate ubiquitination, marking Ptch1 proteins for sorting into endosomes and subsequent degradation, as shown by the previous report². Although this study reported an increase in ciliary Ptch1 levels after Smurf knockdown, this effect is likely a consequence of overall elevated Ptch1 levels in the cytoplasmic membrane, rather than a direct action of Smurf on Ptch1 within the cilium. Indeed, there is no evidence from this study indicating that Smurf localizes to the cilium.

In summary, our results indicate that Numb is a crucial component of the molecular machinery that directly mediates Ptch1 exit from the cilium. Conversely, Smurf proteins are unlikely to be directly involved in Ptch1 exit from the cilium. Instead, Smurf-mediated ubiquitination may operate in conjunction with Numb-mediated Ptch1 endocytosis to determine the destiny of endocytosed Ptch1.

We hope this clarification addresses the reviewer's concern. In this context, we would like to highlight the novelty of our findings and its significance to the cilium study.

- 1) Our study, for the first time, identified an endocytic machinery at the ciliary pocket. Although ciliary pocket has been speculated to be involved in the endocytosis of ciliary proteins, how this is achieved at the molecular level remain unclear. Our study filled this gap by elucidating how the "cargo" molecule Ptch1 is incorporated into the endocytic machinery at the ciliary pocket by Numb. These findings contribute to a broader understanding of modulatory mechanisms governing ciliary levels of receptors and signaling transducers. We suggest that its significance is beyond the regulation of Hh signaling.
- 2) We unraveled part of the molecular mechanism that governs Ptch1 exit from the cilium, a long-standing mystery in Hh signal transduction. Our findings not only pinpointed Numb as an endocytic adaptor for Ptch1, but also highlighted that Ptch1 endocytosis takes place at the ciliary pocket. This process differs essentially from the previously reported effect of Smurf on Ptch1 levels in the cilium.

Overall, we believe that these findings significantly advanced our understanding of the ciliary pocket and opened new avenues for exploring the regulatory mechanism of protein levels in the cilium.

2) A large number of membrane proteins were shown to be removed from cilia upon SHH stimulation (comparing membrane proteins identified from + SHH + biotin verse -SHH + biotin group), however the impact of this interesting finding was not addressed. Further discussion/investigation on the differences among these two groups of proteins and whether other HH signaling-related membrane proteins are impacted in Numb mutant cells, either by immune-staining or proximal labeling, would strengthen the paper.

Response: We thank the reviewer for pointing this out. We are also excited about the other candidate proteins leaving the cilium in response to Shh stimulation. However, these proteins may

not be relevant to Numb and ciliary pocket, as many mechanisms contribute to the exit of protein from the cilium, such as intraflagellar transport (IFT). We believe that the in-depth exploration of these protein candidates belongs to future studies.

(2) The new analyses of Numb ko in ESC-derived NPCs and GCPs in culture strengthen the mechanistic aspect of part by providing some relevance for normal cell types in their in vivo setting. Analysis of in vivo cells (neural tube and GCPs or limb mesenchyme) as previously requested would greatly strengthen the study. Furthermore, if Numb is indeed efficiently removed in conditional ko mutants from GCPs at P6 (in Figure 8 d, f), then the mild cerebellar phenotype indicates a Numb function with other proteins in activating SHH signaling.

Response: We thank the reviewer for acknowledging that our new results from ESC-derived NPCs “strengthen the mechanistic aspect” of our study in *in vivo* relevant settings. We appreciate the reviewer’s suggestions on testing other mouse tissues. However, this would involve a significant amount of time to cross the mouse lines in order to generate the required conditional and homozygous mice, in addition to the long analyses that this would entail. We trust that our results in ESC-derived NPCs, which demonstrate a new cellular type where Numb functions in Shh signaling, effectively address the reviewer’s concerns regarding the relevance of our findings.

For the cerebellar phenotypes in Numb mutants, we would like to point out that the reduced GNP proliferation aligns with the mechanistic roles of Numb in Hh signaling that we identified in cultured cell lines. Numb loss does not completely abolish Hh signaling but rather attenuate the maximal amplitude of the signaling. Our results in NPCs are consistent with this mechanism. Numb loss hinders the differentiation of NPCs into cell fate dependent on high Hh signaling, while showing moderate to no impact on cell fates reliant on low Hh signaling. Therefore, we believe that the GNP proliferation phenotype aligns with the mechanistic roles of Numb in Hh signaling, although it is also possible that Numb may regulate other aspects of cerebellar development via mechanisms beyond Hh signaling.

(5) The result showing SMO is present in Numb ko cells is convincing. The presence of SMO and PTCH1 in Numb mutant cilia is controversial but intriguing.

Response: We thank the reviewer for the comments. It is worth noting that Smo may accumulate in the cilium in an inactive conformation³.

(11) The lobule information should be labeled in the image and/or listed in the legend of Figure 8 d and h. Supplementary Figure 15c H&E control image the tissue is broken, high quality data should be shown.

Response: We thank the reviewer for the suggestions. Unfortunately, we do not have a better image to replace Supplementary Figure 15c. The H&E staining procedure is harsh on frozen sections (especially on large, adult tissues such as the cerebellum) and sometimes results in tissue breaks. We would need to generate new control crosses and harvest new adult tissues, which would take longer than the time that we were given for this last round of revisions. As requested by the reviewer, we have added the lobule information to the images.

Questions from newly added data:

(1) Since Numb acts at the cilium pocket and as an endocytic adaptor to remove Ptch1, in cell types such as ESCs and MEFs that lack a cilium pocket would this role be diminished or replaced by other factors? This should be discussed.

Response: We appreciate the reviewer's question. We speculate that in cells lacking ciliary pocket, endocytosis may take place at the cytoplasmic membrane surrounding the base of the cilium. To support this idea, we examined Numb's localization in IMCD3 cells, a cell line with reported low frequency of ciliary pocket⁴. We infected IMCD3 cells with lentiviruses expressing either Numb-mCherry or Numb-HA, and co-stained cells with a cilium marker. Our results revealed the presence of Numb immunofluorescence signal at the base of the cilium in most transfected cells (Author Response Figure 1). It is possible that Numb proteins in this transition domain between the cilium membrane and the cytoplasmic membrane may carry on similar roles as Numb at the ciliary pocket. We have added a short mention of this in the Discussion (Line # 545-547).

Author Response Figure 1: Subcellular localization of Numb in IMCD3 cells. Numb-mCherry or Numb-HA were expressed via lentiviruses and cells were co-stained with the cilium marker Arl13b. Arrows point to the cytoplasmic membrane surrounding the base of the cilium.

(2) Since for technical reasons the authors cannot detect endogenous Numb with a Numb antibody, they used tagged NUMB protein to stain ciliary compartments. The authors should provide quantification of the % of cells that have NUMB (Numb-V5 or -GFP) localized to the cilium pocket/base of total cells with tagged protein. This is important because in Figure 3h and supplemental Figure 6a, the Numb-V5 channel shows no cilia base staining. Also to address is the Numb localization to the cilium pocket related to the level of SHH signaling?

Response: We appreciate the reviewer's question. The quantification result of the percentage of cells containing Numb-full length at the cilium was provided in our previous version (33% of cells) (Line #219). We apologize for not highlighting this information to the reviewer's attention.

In this round of revision, we quantified the percentage of cilium localization for all truncated Numb proteins generated in our study and included the results in Supplementary Figure 5a. These results provide further insights into the dynamic nature of Numb's localization to the ciliary pocket. Notably, the truncated construct NPTB exhibits the highest percentage of localization at the cilium (86.4%). This is likely attributed to the absence of C-terminal motifs, such as the NPF and DPF domains known to bind to adaptins⁵⁻⁹. As a result, NPTB cannot be incorporated into the clathrin-coated vesicles and tends to remain "stuck" at the ciliary pocket. Meanwhile, this result suggests that the full-length Numb protein localizes to the ciliary pocket in a highly dynamic manner: Numb is integrated into the endocytosis machinery and exit the ciliary pocket immediately after it reaches

the pocket. This could also account for our observation of Numb being present in the cilia in only 33% of cells. We incorporated this analysis to the Discussion section (Line # 551-558).

Due to this transient and dynamic nature of Numb's localization to the ciliary pocket, there is no apparent correlation between Numb's ciliary localization and Hh signaling. In fact, the NPTB construct exhibits the highest frequency of localization to the cilium but fails to rescue Hh signaling in Numb-loss cells (Figure S9d), likely because NPTB is unable to mediate Ptch1 endocytosis.

Reviewer #3 (Remarks to the Author):

Thank you for addressing all of my minor criticisms of the original submission. All of my concerns are addressed now.

Thank you for your constructive comments that helped improve our manuscript.

Reviewer #4 (Remarks to the Author):

the authors addressed my concerns fully.

Thank you and we appreciate your constructive comments that helped improve our manuscript.

References:

1. Dho, S.E., French, M.B., Woods, S.A. & McGlade, C.J. Characterization of Four Mammalian Numb Protein Isoforms. *Journal of Biological Chemistry* **274**, 33097-33104 (1999).
2. Yue, S. *et al.* Requirement of Smurf-mediated endocytosis of Patched1 in sonic hedgehog signal reception. *eLife* **3**, 1-24 (2014).
3. Rohatgi, R., Milenkovic, L., Corcoran, R.B. & Scott, M.P. Hedgehog signal transduction by Smoothed: Pharmacologic evidence for a 2-step activation process. *Proceedings of the National Academy of Sciences* **106**, 3196-3201 (2009).
4. Molla-Herman, A. *et al.* The ciliary pocket: an endocytic membrane domain at the base of primary and motile cilia. *J Cell Sci* **123**, 1785-1795 (2010).
5. Salcini, A.E. *et al.* Binding specificity and in vivo targets of the EH domain, a novel protein-protein interaction module. *Genes Dev* **11**, 2239-2249 (1997).
6. Smith, C.A., Dho, S.E., Donaldson, J., Tepass, U. & McGlade, C.J. The cell fate determinant numb interacts with EHD/Rme-1 family proteins and has a role in endocytic recycling. *Mol Biol Cell* **15**, 3698-3708 (2004).
7. Berdnik, D., Török, T., González-Gaitán, M. & Knoblich, J.A. The Endocytic Protein α -Adaptin Is Required for Numb-Mediated Asymmetric Cell Division in *Drosophila*. *Developmental Cell* **3**, 221-231 (2002).
8. Santolini, E. *et al.* Numb is an endocytic protein. *J Cell Biol* **151**, 1345-1352 (2000).
9. Song, Y. & Lu, B. Interaction of Notch signaling modulator Numb with alpha-Adaptin regulates endocytosis of Notch pathway components and cell fate determination of neural stem cells. *J Biol Chem* **287**, 17716-17728 (2012).

REVIEWERS' COMMENTS

Reviewer #1 (Remarks to the Author):

The authors have addressed my previous concerns.

Reviewer #2 (Remarks to the Author):

For the reason of time, these suggested experiments involving in vivo Neural tube patterning data and better H&E staining of adult cerebellum, fail to be fulfilled. Otherwise, other concerns are well addressed in the revised version.